# Combined physical and pharmacological anabolic osteoporosis therapies increase bone response and mechanoregulation in female mice

Friederike A. Schulte[1], Francisco C. Marques [1], Julia K. Griesbach[1], Claudia Weigt[1], Marcella von Salis-Soglio[1], Floor M. Lambers[1], Clemens Kreutz [2], Michaela Kneissel[2], Peter J. Richards[2,3], Gisela A. Kuhn[1] & Ralph Müller [1] ✉

Bone's ability to adapt to mechanical demands is governed by mechanoregulation, the process by which cells sense and respond to mechanical stimuli to maintain skeletal integrity. In osteoporosis, increased bone resorption activity leads to structural deterioration and elevated fracture risk. While existing pharmacological therapies aim to restore bone mass to reduce fracture risk, it is unclear how they modulate mechanoregulation, especially when combined with physical interventions. Here, we investigate the joint effects of load-bearing physical and pharmacological treatment in a female mouse model of osteoporosis using longitudinal in vivo micro-computed tomography and computational mechanics. We demonstrate that mechanical loading additively and synergistically enhanced predicted strength, bone volume, and mechanoregulation parameters when combined with anabolic therapies (parathyroid hormone and sclerostin antibody) but not with anti-catabolic treatments (bisphosphonates). Increases in predicted strength are associated with reductions in bone resorption rates, shifts in the (re)modeling thresholds as anticipated by Frost in the mechanostat theory, and the modeling capacity of anabolic pharmacological treatments. These findings underscore the therapeutic potential of combining anabolic pharmacological therapies with load-bearing physical activity, particularly in early treatment phases, to optimize bone adaptation and fracture prevention in osteoporosis management.

Postmenopausal osteoporosis in women is characterized by increased bone resorption, leading to structural deterioration of the bone microarchitecture and increased susceptibility to fractures[1]. Exercise is fundamental to bone health, but its role alongside pharmacological treatment remains an area of active inquiry[2]. Patients with diagnosed osteoporosis often question the compatibility of exercise with pharmacological treatment, some fearing interference with drug efficacy, others assuming that their potent medications obviate the need for

---

[1]Institute for Biomechanics, ETH Zurich, Zurich, Switzerland. [2]Novartis Biomedical Research, Novartis Campus, Basel, Switzerland. [3]Bone and Stem Cell Research Group, CABMM, University of Zurich, Zurich, Switzerland. ✉e-mail: ram@ethz.ch

physical activity. This highlights the need for evidence on whether mechanical loading improves outcomes for bone strength beyond pharmacological monotherapy.

Mechanical loads are transmitted from the organ to bone cells, where osteocytes act as mechanosensors, osteoblasts form, and osteoclasts resorb bone. This interplay drives both remodeling (resorption followed by formation) and modeling (formation without prior resorption), the latter potentially via lining cell redifferentiation in the absence of sclerostin[3]. Key signaling pathways between bone cells include Wnt/β-catenin[4] (osteocyte-osteoblast) and RANKL/OPG[5] (osteoblast-osteoclast). To study these dynamic processes in vivo, high-resolution imaging techniques, such as micro-computed tomography (micro-CT) are commonly used. While micro-CT does not resolve individual cells or molecular signals, it captures the net microstructural changes resulting from formation and resorption, which are collectively referred to as bone (re)modeling[6].

On the cellular level, anti-resorptive bisphosphonates (BIS) prevent osteoclast attachment and activity by binding to hydroxyapatite crystals in the bone matrix, and are characterized by low osteoanabolic capacity[7]. We hypothesized that mechanical loading would stimulate bone formation, but bisphosphonates might blunt this response by inhibiting (re)modeling.

Intermittent parathyroid hormone (PTH) exerts anabolic effects by binding to PTH-receptors on osteoblasts and osteocytes, leading to suppression of sclerostin expression and subsequent activation of the Wnt/β-catenin signaling pathway. This promotes osteoblastic activity and indirectly enhances osteoclast recruitment via upregulation of RANKL expression by osteoblasts[8]. PTH and mechanical loading have shown synergistic effects in cortical bone[9,10]. We hypothesized that in trabecular bone, their combination would suppress otherwise increased resorption and concurrently stimulate bone formation[11].

Sclerostin antibodies (SclAB) inhibit a Wnt/β-catenin signaling antagonist, promoting osteoblast activity and suppressing osteoclast activity[12]. Wnt/β-catenin-signaling is initiated when a Wnt ligand binds to a Frizzled family receptor and its co-receptor LRP5/6 on the surface of a cell, resulting in β-catenin stabilization and target gene transcription inside the cell[11]. We hypothesized that mechanical loading, in combination with SclAB, would further increase formation and decrease resorption. Synergy may arise because mechanical loading downregulates sclerostin expression in osteocytes[13], while SclAB pharmacologically neutralizes it, amplifying the same pathway from two directions, especially in trabecular bone, which particularly adapts its micro-architecture to mechanical demands. It should be noted that, similar to mechanical loading, sclerostin antibody effects saturate after a certain time, which limits their long-term use as a standalone therapy[14].

Frost's mechanostat theory[15-17] proposes that bone adapts to maintain mechanical strain within a defined setpoint range - much like a thermostat. The setpoints separate the following four states: the disuse state for bone resorption; the adapted state for quiescence; the overload state for formation; and the fracture state for microdamage and fractures. The mechanostat theory was introduced more than 30 years ago. In this theoretical construct, anti-catabolic bisphosphonates help inhibit increased bone resorption in underloaded areas by raising the mechanical setpoints towards higher strains. Anabolic treatments theoretically promote bone formation by lowering the mechanical setpoints required to initiate formation, provided mechanical loading is also encouraged. Measuring the "setpoints" of osteoporosis medications in the context of Frost's mechanostat theory, however, is not straightforward, as the setpoints themselves are theoretical constructs that describe the mechanical thresholds at which bones adjust their structure, and measuring them experimentally would require complex assessments of both bone volume and mechanical strain under highly controlled conditions.

Here, we provide experimental insights into this interaction using in vivo micro-computed tomography (micro-CT) and finite-element modeling in a C57BL/6 mouse model of osteoporosis. The aim of this study was to explore whether combining pharmacological treatments with mechanical loading improves pharmacological treatment outcomes, i.e., stronger bones. We tested the short-term effects of BIS, PTH, and SclAB treatment with or without mechanical loading of the 6th caudal vertebra, aiming to maximize bone accrual in early treatment phases. The choice of selected therapies was based on clinical relevance and distinct modes of action in the bone (re)modeling process.

## Results

### Micro-architectural and (re)modeling differences reveal treatment-specific effects on bone recovery after ovariectomy

As shown in Fig. 1a, C57BL/6 mice were ovariectomized at 15 weeks of age, and bone loss was confirmed in all animals ($n = 105$) via in vivo micro-CT of the 6th caudal vertebra (-26.8 ± 0.95% change in bone volume fraction (BV/TV) from baseline). From week 20 to 24, animals received pharmacological and/or mechanical treatment through an established tail vertebra loading model[18].

In the vehicle (VEH)-group, BV/TV continued to decline from week 20, while all monotreatment groups showed significant increases after four weeks of treatment ($p < 0.001$, Fig. 1b). Trabecular thickness (Tb.Th) did not increase significantly in BIS compared to VEH, whereas all anabolic treatments led to increases, but lower in PTH than in mechanical loading (ML) and SclAB ($p < 0.01$, Fig. 1c). Trabecular number (Tb.N) was preserved in BIS but was significantly lower in all other groups compared to BIS ($p < 0.001$, Fig. 1d). Connectivity density (Conn.D) was preserved in BIS and PTH and declined in VEH, ML, and SclAB ($p < 0.05$, Fig. 1e). A full summary of these microstructural parameters is provided in Supplementary Dataset S.2. As previously described[18], mechanical loading increases BV/TV mainly through trabecular thickening despite a reduction in trabecular number, explainable by a full loss of thin trabeculae - a pattern also seen in the SclAB-group (Fig. 1c, d). In contrast, PTH increased BV/TV through moderate thickening and sustained preservation of trabecular number, resulting in a smaller loss of connectivity. BIS maintained BV/TV by preserving existing trabeculae, including thinner ones, with minimal thickening (Fig. 1, b-e).

The micro-architectural changes resulted from the time course of formation and resorption (Fig. 1f). In week interval w20-22 (see also Supplementary Dataset S.5 for results from the second interval, w22-24), bone formation rate per bone volume (BFR) was significantly increased in ML and PTH compared to VEH, with no significant increases in BIS or SclAB (Fig. 1g). Mineral apposition rate (MAR) did not differ between groups (Fig. 1h), while mineralizing surface per bone surface (MS) was elevated in all treatment groups compared to VEH ($p < 0.05$, Fig. 1i). Bone resorption rate per bone volume (BRR) was significantly reduced in all treatment groups versus VEH (Fig. 1j), where BRR in PTH was the least reduced (significantly higher than VEH, ML, and BIS), consistent with the known anabolic effects of intermittent PTH treatment. Surprisingly, BRR was significantly lower than VEH, but not as low as in ML and BIS. This decrease, even more pronounced in the second time interval (see Supplementary Dataset S.5), may result from micro-CT missing parts of the reversal phase between resorption and formation, thus potentially underrepresenting remodeling at the same sites. As expected, mineral resorption rate (MRR) was significantly lower in the BIS-group compared to VEH and unchanged in the anabolic groups (Fig. 1k), suggesting bisphosphonates preserve Tb.N (Fig. 1d) by preventing full trabecular resorption. Eroded surface per bone surface (ES), reflecting the proportion of bone surface previously affected by resorption, was significantly lower in ML and SclAB compared to VEH, BIS, and PTH ($p < 0.05$, Fig. 1l).

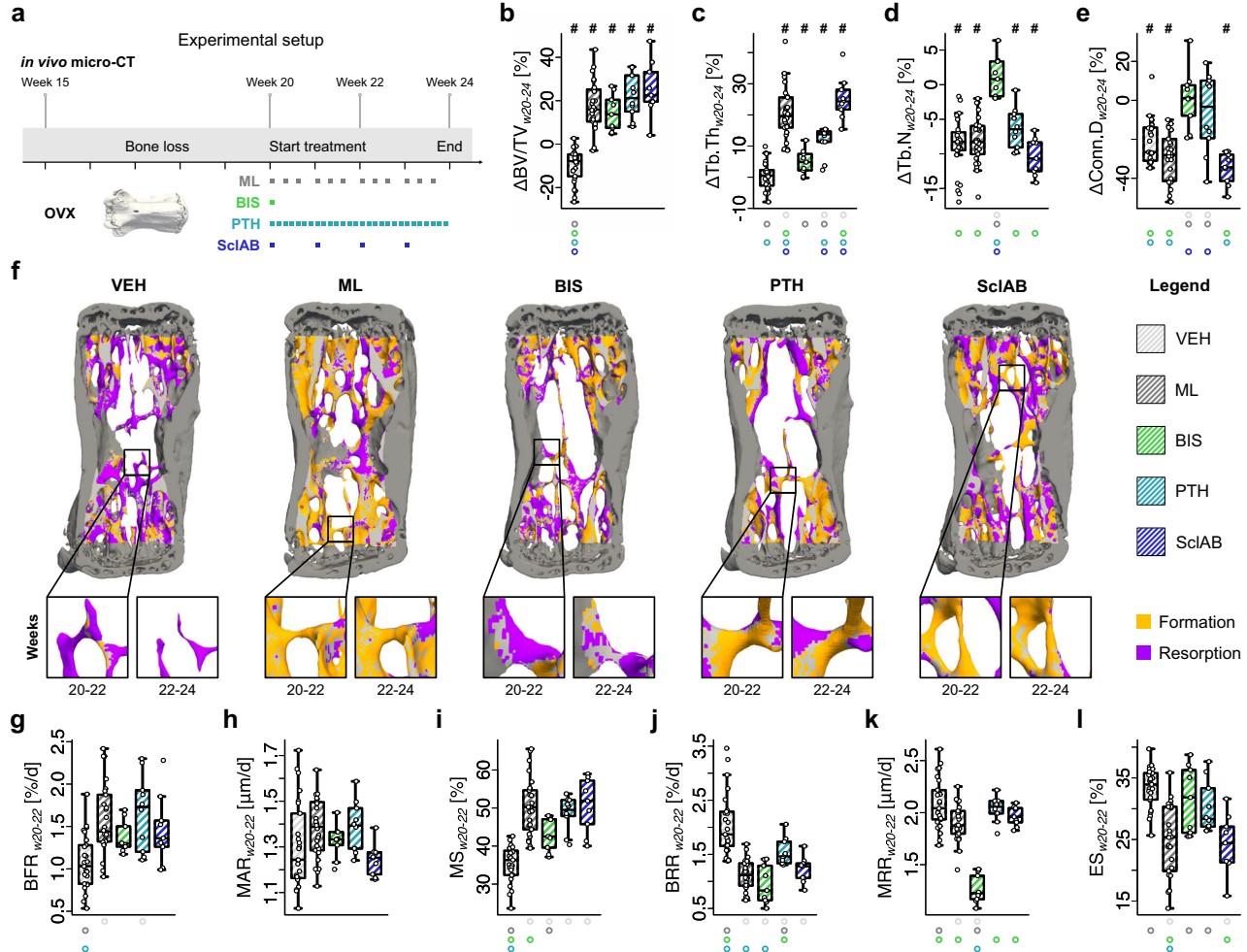

**Fig. 1 | Osteoporosis monotreatments enhance trabecular bone micro-architecture by different mechanisms of action. a** Experimental setup. 15-week-old mice are ovariectomized, and treatment is administered between week 20 (w20) and 24 (w24). (**b**–**e**, **g**–**l**) Monotreatments (VEH: $n = 25$, BIS: $n = 9$, PTH: $n = 10$, SclAB: $n = 9$; each data point represents one animal (biological replicate)), boxplots show percentage change. The box spans the interquartile range (IQR) from the 25th to the 75th percentile. The median is shown as a line inside the box. Whiskers extend to the most extreme data points within 1.5 × IQR, points beyond this range are plotted as outliers. # indicates that the delta of the group is different from zero, as tested with t-test with null-hypothesis. **b** bone volume fraction ΔBV/TV,

**c** trabecular thickness ΔTb.Th, **d** trabecular number ΔTb.N **e** connectivity density ΔConn.D. **f** Representative micro-computed tomography overlays at intervals w20-22 and w22-24, with formation and resorption volumes **g**–**l** (Re)modeling rates calculated from time interval w20-22 **g** bone formation rate per bone volume BFR **h** mineral apposition rate MAR **i** mineralizing surface per bone surface MS **j** bone resorption rate per bone volume BRR **k** mineral resorption rate MRR **l** eroded surface per bone surface ES. Significant differences are depicted with a symbol in the color of the corresponding group (○: to monotreatment, ●: to combined treatment, see Supplementary Dataset S.1 for color encoding, $p < 0.05$). Source data are provided as a Source Data file.

---

Together, these results confirm that each monotherapy exerted effects consistent with its known mode of action: bisphosphonates preserving trabecular structure by suppressing bone resorption, and anabolic agents increasing bone mass predominantly through formation, while also highlighting treatment-specific differences in trabecular architecture and dynamic turnover.

### Mechanical loading modulates structural and functional responses to pharmacological interventions

We next assessed how additional mechanical loading influenced the response to pharmacological treatment. In the BIS + ML-group, BV/TV gains were significantly less than the expected additive effects (treatment:loading: $p < 0.05$, Fig. 2a), and did not differ significantly from BIS alone (Supplementary Dataset S.2). In contrast, anabolic treatment combined with mechanical loading (PTH + ML, SclAB + ML) resulted in additive increases, as supported by a non-significant treatment:loading interaction (Fig. 2a). ΔTb.Th was significantly higher in all combination groups compared to monotreatments after four weeks (Fig. 2b). ΔTb.N

was preserved in BIS + ML, despite the known Tb.N loss with ML alone (Fig. 2c), suggesting BIS blunted this effect. The loss of Tb.N observed in SclAB was not rescued in SclAB + ML (Fig. 2c). Conn.D as a measure of the structural integrity and interconnectedness of the trabecular network, was preserved the most in BIS + ML, and PTH + ML (Fig. 2d). It should be noted that structural parameters, such as Tb.Th and Tb.N, are interdependent; apparent increases in Tb.Th may arise from the loss of complete thinner trabeculae. Synergy was found in total area (Tt.Ar), leading to additive increases in cortical area fraction (Ct.Ar/Tt.Ar) and cortical thickness (Ct.Th), found in Supplementary Fig. S.1 and Dataset S.4. Predicted strength increased in all treatment groups, with loading-induced increases being additive in the PTH + ML group and synergistic in the SclAB + ML group ($p < 0.05$, Fig. 2e). Strain energy density (SED) decreased in all treatment groups, being most pronounced in the combination groups, with less than the expected additive reduction in BIS + ML (treatment:loading: $p < 0.05$, Fig. 2f). Illustrating the effect of decreasing SED, the visualization of the 6th caudal vertebra in a SclAB + ML animal shows a highly loaded trabecula

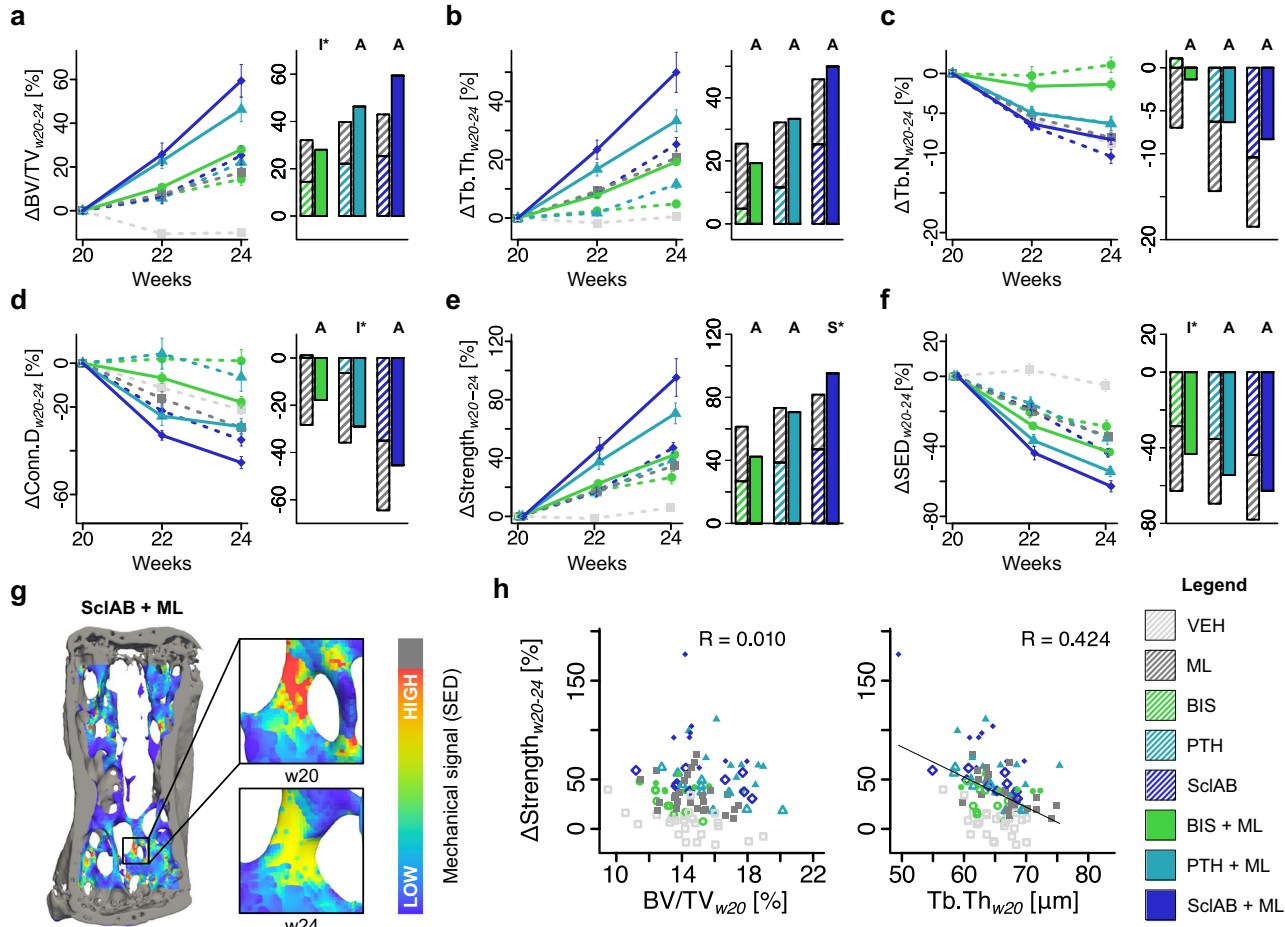

**Fig. 2 | Mechanical loading increases the morphological effects of anabolic and anti-catabolic treatments, contributing to increased predicted strength.** **a–f** Changes from baseline. Data are presented as mean ± standard error. Each data point represents one animal (biological replicate; VEH: $n = 25$, ML: $n = 26$, BIS: $n = 9$, BIS + ML: $n = 9$, PTH: $n = 10$, PTH + ML: $n = 9$, SclAB: $n = 9$, SclAB + ML: $n = 8$) **a** bone volume fraction (ΔBV/TV) **b** trabecular thickness (ΔTb.Th) **c** trabecular number (ΔTb.N) **d** connectivity density (ΔConn.D) **e** predicted strength (ΔStrength), and **f** strain energy density (ΔSED). Bars show group means of sum of single vs. combined treatment indicating: antagonistic (I), additive (A) or synergistic (S) effects determined via treatment:loading interaction in a linear model on the %-change. **g** Example of SED distribution in the 6th caudal vertebra of a SclAB + ML animal at week 20, showing reduced SED after structural reorganization from week 20 to 24. **h** Linear regression of ΔStrength with BV/TV and Tb.Th at week 20. Source data are provided as a Source Data file.

at week 20, followed by structural reorganization and reduced SED at week 24 (see Fig. 2g). Linear regression analysis revealed that the changes in predicted strength did not correlate with baseline BV/TV ($R = 0.010$) and moderately with Tb.Th ($R = 0.424$, Fig. 2h), indicating that initial BV/TV or Tb.Th serve as poor or only moderate predictors of future changes in strength.

## Combination treatments affect bone formation and resorption dynamics differentially

BFR was lowest in VEH and highest in the anabolic combination groups ($p < 0.05$, Fig. 3a). MAR showed synergy in SclAB + ML (treatment:loading: $p < 0.05$, Fig. 3b). MS increased additively in PTH + ML and SclAB + ML, and antagonistically in BIS + ML (treatment:loading: $p < 0.01$, Fig. 3c). BRR was reduced in all treatment groups ($p < 0.05$, Fig. 3d). In BIS + ML, the BRR reduction was antagonistically affected by mechanical loading (treatment:loading, $p < 0.05$). In contrast, BRR decreased synergistically in PTH + ML (treatment:loading: $p < 0.05$, Fig. 3d) and additively in SclAB + ML (Fig. 3d). MRR remained suppressed in BIS + ML and was unaffected by mechanical loading (Fig. 3e). ES decreased synergistically in PTH + ML (treatment:loading: $p < 0.05$, Fig. 3f) and SclAB + ML (treatment:loading: $p < 0.01$, Fig. 3f). This synergistic effect was, at least in SclAB + ML, still present in the second time interval (see Supplementary Fig. S.2 and Dataset S.5 for

results from w22–24). Representative trabecular microstructures and insets of the combination groups at both time intervals are depicted in Fig. 3g. Changes in predicted strength correlated linearly with BFR ($R = 0.830$) and decayed according to a power-law relationship with BRR ($\rho = 0.760$), indicating that small reductions in bone resorption rates under treatment can result in large increases in predicted strength (Fig. 3h).

## Bone formation and resorption occur in mechanically distinct strain windows

To address the question whether pharmacological treatments target bone (re)modeling to the places that mechanically need it, we correlated the underlying mechanical signals in the trabecular compartment with subsequent formation and resorption events. Visually, spatial agreement was found between high SED and regions of subsequent bone formation, and low SED and regions of subsequent bone resorption (Fig. 4a). Quantitative analysis (Fig. 4b) showed that mean SED was significantly higher at formation sites ($p < 0.05$) and significantly lower at resorption sites ($p < 0.05$) compared to quiescent surfaces, when presented as percentage change to the average quiescent SED across all animals per group. The addition of mechanical loading to the monotreatments resulted in marginal changes, indicating that bone formation and resorption continue to occur in

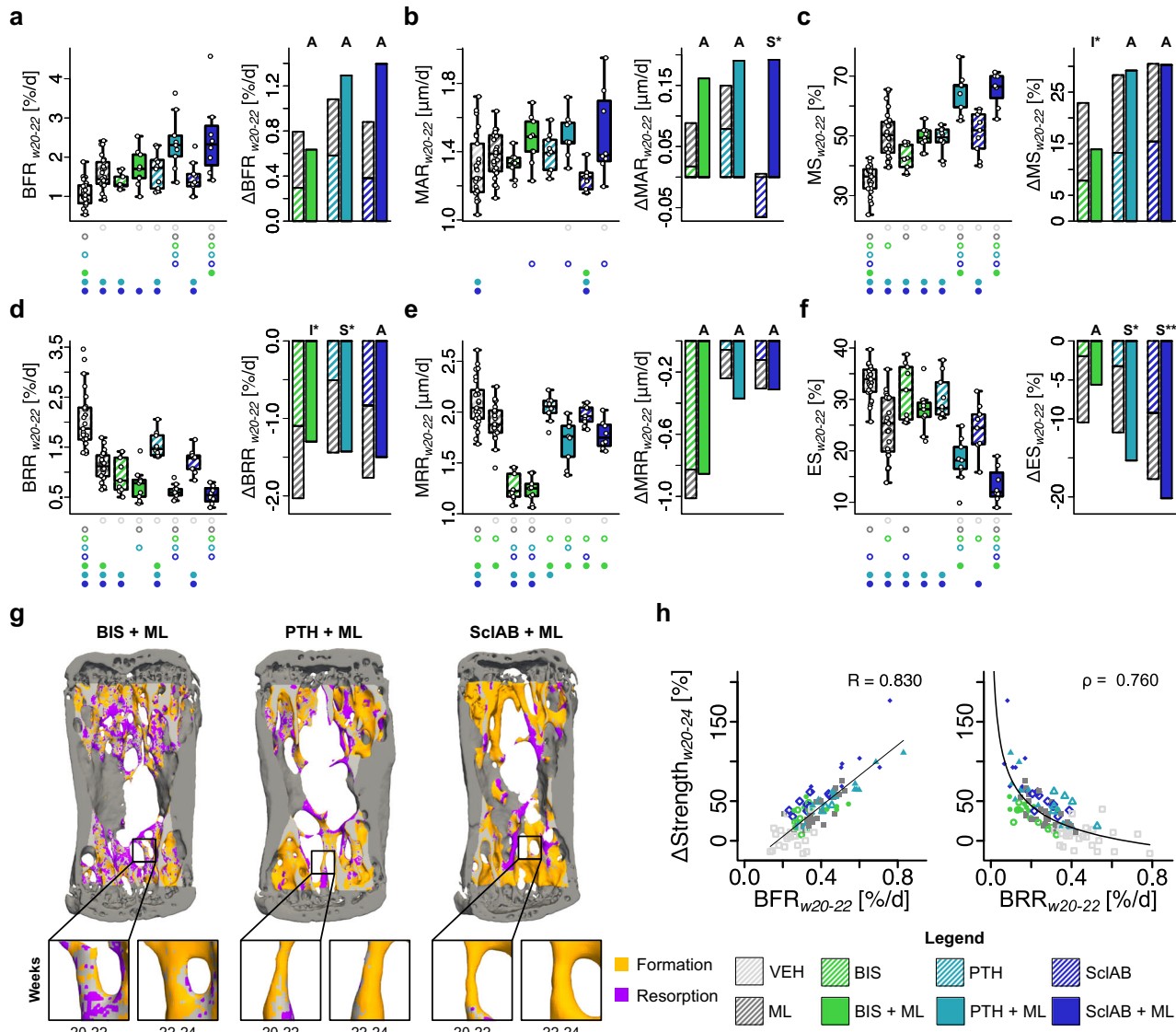

**Fig. 3 | Mechanical loading increases bone formation and decreases resorption rates of anabolic and anti-catabolic pharmacological treatments which contribute to increased predicted strength. a** bone formation rate per bone volume (BFR) **b** mineral apposition rate (MAR) **c** mineralizing surface per bone surface (MS) **d** bone resorption rate per bone volume (BRR) **e** mineral resorption rate (MRR) **f** eroded surface per bone surface (ES). **a–f** Each data point represents one animal (biological replicate; VEH: $n = 25$, ML: $n = 26$, BIS: $n = 9$, BIS + ML: $n = 9$, PTH: $n = 10$, PTH + ML: $n = 9$, SclAB: $n = 9$, SclAB + ML: $n = 8$). The box spans the interquartile range (IQR) from the 25th to the 75th percentile. The median is shown as a line inside the box. Whiskers extend to the most extreme data points within $1.5 \times$ IQR, points beyond this range are plotted as outliers. Δ denotes difference from VEH as

no baseline value for dynamic bone morphometry rates was available. Bars show group means of sum of single vs. combined treatment indicating: antagonistic (I), additive (A) or synergistic (S) effects determined via treatment:loading interaction in a linear model on the $\log_2$-transformed values. Significances were calculated using one-way ANOVA followed by Tukey's post-hoc test; significant differences are depicted with a symbol in the color of the corresponding group (o: to mono-treatment, ●: to combined treatment, see Supplementary Dataset S.1 for color encoding, $p < 0.05$). **g** Visual representations of combination groups BIS + ML, PTH + ML, and SclAB + ML in week intervals w20-22 and w22-24 **h** Linear regression of ΔStrength with BFR, and power-law regression with BRR in week interval w20-22. Source data are provided as a Source Data file.

mechanically distinct strain windows despite possible influences of pharmacological interventions. Levene's test revealed significantly smaller variances between BIS and BIS + ML and between VEH and ML ($p < 0.05$), indicative of more consistent mechanosensitive responses in these groups in the presence of mechanical loading.

Conditional probabilities of formation, quiescence, and resorption events for a given SED-value were calculated and fitted with exponential functions. The probability for bone formation increased with increasing SED, and the probability for bone resorption decreased with increasing SED (Fig. 4c, Supplementary Fig. S.3 and Dataset S.6 for results in w22-24). Especially the PTH + ML and SclAB + ML conditional probability curves opened up compared to their monotreatments,

indicating a higher responsiveness to mechanical signals (Fig. 4c). The asymptotic resorption probability value $min_R$ represents the point at which further increases in local mechanical loading stimulus do not further suppress bone resorption. This value was significantly lower in ML, PTH + ML, and in SclAB + ML than in VEH ($p < 0.05$, Fig. 4d, Supplementary Dataset S.8). Further comparisons of the coefficients from fitted functions to the conditional probabilities are provided in Supplementary Dataset S.7 and S.8. The presence of few significant differences between groups suggests mechanosensitivity of the system is largely preserved regardless of the presence or absence of mechanical loading. Combined anabolic treatments resulted in higher correct classification rate (CCR) values compared to VEH and the BIS groups

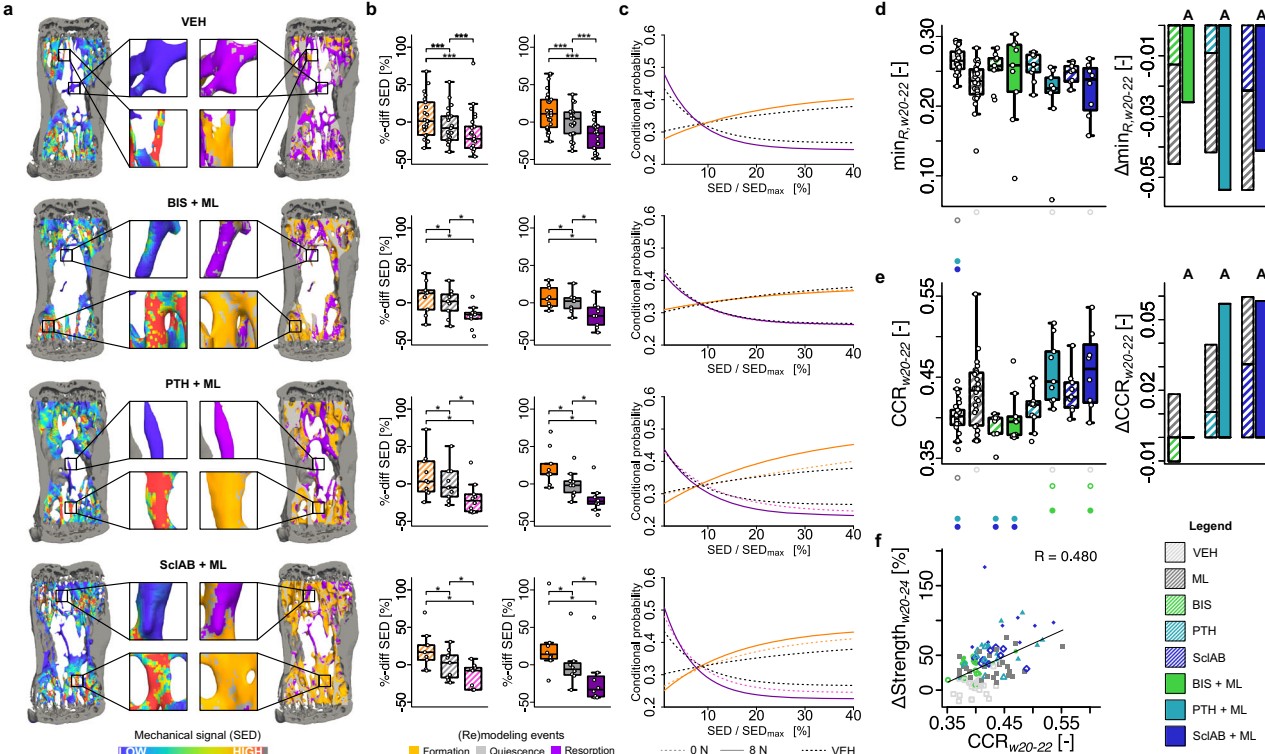

**Fig. 4 | Formation aligns with high mechanical signal, resorption with low.**
**a** Spatial comparison of local mechanical signal (SED at week 20) with sites of trabecular bone formation and resorption (w20-22). **b** SED at formation, resorption, and quiescent sites[6], normalized to the group mean of quiescence to allow statistical testing with this group. Each data point represents one animal (biological replicate; VEH: $n = 25$, ML: $n = 26$, BIS: $n = 9$, BIS + ML: $n = 9$, PTH: $n = 10$, PTH + ML: $n = 9$, SclAB: $n = 9$, SclAB + ML: $n = 8$). The box spans the interquartile range (IQR) from the 25th to the 75th percentile. The median is shown as a line inside the box. Whiskers extend to the most extreme data points within $1.5 \times$ IQR, points beyond this range are plotted as outliers. *$p < 0.05$, ***$<0.001$ using two-sided paired Wilcoxon-signed rank test with Bonferroni correction; exact $p$-values are found in Supplementary Dataset S.6. **c** The plots show the exponential fitting functions (taking all animals per group into account) for conditional probability of formation and resorption as a function of $SED/SED_{max}$. Dashed lines represent monotreatment (0 N, including VEH), and solid lines represent combined treatment with mechanical loading (8 N, including ML). VEH is indicated by black dotted line.

Values are cropped above 40% due to low voxel count[6]. **d,e** Each data point represents one animal (biological replicate; VEH: $n = 25$, ML: $n = 26$, BIS: $n = 9$, BIS + ML: $n = 9$, PTH: $n = 10$, PTH + ML: $n = 9$, SclAB: $n = 9$, SclAB + ML: $n = 8$). The box spans the interquartile range (IQR) from the 25th to the 75th percentile. The median is shown as a line inside the box. Whiskers extend to the most extreme data points within $1.5 \times$ IQR, points beyond this range are plotted as outliers. [-] denotes unitless, $\Delta$ denotes change from VEH; sum of single vs. combined treatment group means indicating antagonistic (I), additive (A) or synergistic (S) effects.
**d** Asymptotic resorption probability value ($min_R$). **e** CCR as a measure of mechanoregulated (re)modeling, with the $\Delta$CCR bar for the BIS + ML group close to zero, making it barely visible. **f** Linear regression analysis in week interval w20-22 of $\Delta$Strength with CCR. Significant differences are depicted with a symbol in the color of the corresponding group (o: to monotreatment, ●: to combined treatment, see Supplementary Dataset S.1 for color encoding, $p < 0.05$). Source data are provided as a Source Data file.

(Fig. 4e), with CCR being a metric used to evaluate the accuracy of classifying local bone formation or resorption events based on predicted mechanical signals. However, CCR only correlated moderately with the increases in strength, indicating that increased mechanosensitivity of the system may not fully explain the ability of bone to increase strength (Fig. 4f).

**Frost's mechanostat setpoints are shifted towards lower values for mono- and combined treatments**
To assess potential shifts in the setpoints due to different drug or combination treatments, (re)modeling velocity (RmV) curves and mechanostat parameters[19] were calculated for each treatment group as an approximation to the mechanostat theory. These curves show the net RmV for a given surface effective strain, where a negative value means net resorption and a positive value means net formation (Fig. 5a). From these curves, formation saturation levels (FSL), resorption saturation levels (RSL), and (re)modeling thresholds (RmTs) were derived.

FSL was increased in all treatments compared to VEH ($p < 0.001$, Fig. 5b, Supplementary Dataset S.9), and was significantly higher in

anabolic combination groups compared to the respective monotreatments ($p < 0.001$). This suggests an increased mechanosensitivity of the cells involved in bone formation. RSL was increased in the BIS and all combination groups compared to VEH ($p < 0.05$, Fig. 5c). The increase in RSL under bisphosphonate treatment may be explained by the suppression of bone turnover: at comparable strain levels, resorption would typically occur, resulting in a lower RSL; however, due to inhibited osteoclast activity, this resorptive response is blunted, making the system appear saturated at a higher RSL value. Regarding the role of additional mechanical loading, only the PTH + ML group showed a significant increase compared to PTH alone ($p < 0.05$, Supplementary Dataset S.9). This suggests that, under strain conditions that would typically lead to bone resorption, the balance has shifted towards bone formation. RmT values, representing the setpoints in the mechanostat theory, were decreased in all pharmacological treatment groups compared to VEH ($p < 0.05$), with no significant improvements observed in the combination groups receiving mechanical loading (Fig. 5d). These results were even more pronounced in the second time interval (Supplementary Fig. 4d and Dataset S.9). The reduction in RmT suggests an upregulation of osteocytic mechanosensitivity, an

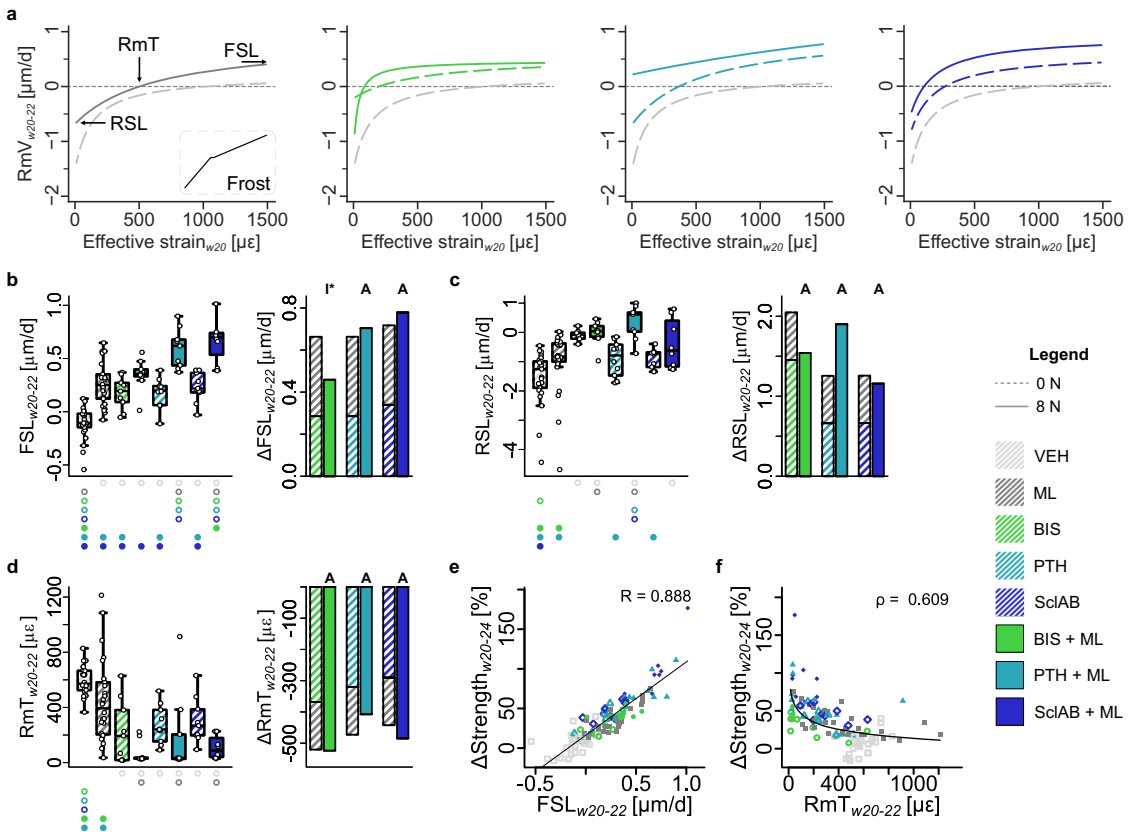

**Fig. 5 | (Re)modeling velocity curves as an approximation of the mechanostat theory. a** (re)modeling velocity (RmV) curve fits (taking all animals per group into account) for VEH / ML, BIS / BIS + ML, PTH / PTH + ML, SclAB / SclAB + ML; **b** formation saturation level (FSL) **c** resorption saturation level (RSL), **d** (re)modeling threshold (RmT) **b**–**d** Each data point represents one animal (biological replicate; VEH: $n = 25$, ML: $n = 26$, BIS: $n = 9$, BIS + ML: $n = 9$, PTH: $n = 10$, PTH + ML: $n = 9$, SclAB: $n = 9$, SclAB + ML: $n = 8$). The box spans the interquartile range (IQR) from the 25th to the 75th percentile. The median is shown as a line inside the box. Whiskers extend to the most extreme data points within 1.5 × IQR, points beyond this range are plotted as outliers. Δ denotes difference from VEH as no baseline

value from week 20 was available. Bars show group means of sum of single vs. combined treatment indicating: antagonistic (I), additive (A) or synergistic (S) effects determined via treatment:loading interaction in a linear model on the absolute values. Significances were calculated using one-way ANOVA followed by Tukey's post-hoc test; significant differences are depicted with a symbol in the color of the corresponding group (o: to monotreatment, ●: to combined treatment, see Supplementary Dataset S.1 for color encoding, $p < 0.05$). **e** Linear regression of ΔStrength with FSL, and **f** power-law regression with RmT from week interval w20-22. Source data are provided as a Source Data file.

expected outcome in the anabolic treatment groups, but an unexpected finding in the bisphosphonate-treated groups. This may reflect a compensatory osteocytic response to the suppressed bone resorption induced by bisphosphonate treatment. Furthermore, FSL correlated strongly with predicted strength increases (R = 0.888), while RmT exhibited a moderate inverse correlation ($\rho = 0.609$) when fitted with a power-law function. These findings suggest that a reduction in RmT may be a prerequisite for strength gains; however, actual increases in strength require anabolic potential (Fig. 5e, f).

### The amount of modeling correlates highly with increases in predicted strength for all treatments

Next, we distinguished between the remodeling and the modeling ability of the different treatments. Visually, in several areas in VEH, high SED values decreased over time, but in other areas they increased and the resorption of a single trabecula led to a more heterogeneous SED distribution (Fig. 6a). While SED on average decreased over time for BIS + ML, this did not particularly affect areas with high SED, as there remained several highly strained areas in week 24. In contrast, both PTH + ML and SclAB + ML showed large decreases in SED, especially in areas with high SED values in week 20 (Fig. 6a).

The remodeling ability - as represented by the amount of resorption followed by formation - was in BIS, BIS + ML, PTH and SclAB

comparable to VEH, and significantly lower in ML and the anabolic combination groups compared to VEH ($p < 0.05$, Fig. 6b). In contrast, the modeling ability was significantly higher compared to VEH in all but the BIS group, with the highest amount in the anabolic combination treatments ($p < 0.001$, Fig. 6c). Both remodeling and modeling capacity showed antagonistic effects in BIS + ML and additive effects in anabolic combination treatments (Fig. 6b, c). Remodeling showed a moderate inverse correlation with increases in predicted strength when fitted with a power-law function ($\rho = 0.476$, Fig. 6d), while modeling correlated strongly in a linear fit (R = 0.928, Fig. 6e). Figure 6f illustrates the distribution of surface events relative to increasing effective strain. Remodeling events appeared strain-independent, and modeling increased with higher mechanical stimuli, particularly in the anabolic combination groups (Fig. 6f). There is also a non-linear increase in modeling with mechanical loading in comparison to the mono-treatment, with a stronger effect for low effective strain regions, especially for SclAB. Figure 6g summarizes the trabecular bone response across treatments. Bisphosphonates, alone or with mechanical loading, help preserve trabecular interconnectedness. Anabolic treatments combined with mechanical loading effectively enhance predicted strength by promoting formation and suppressing resorption. These additive effects across parameters likely underlie the observed synergistic gains in predicted strength, at least in SclAB.

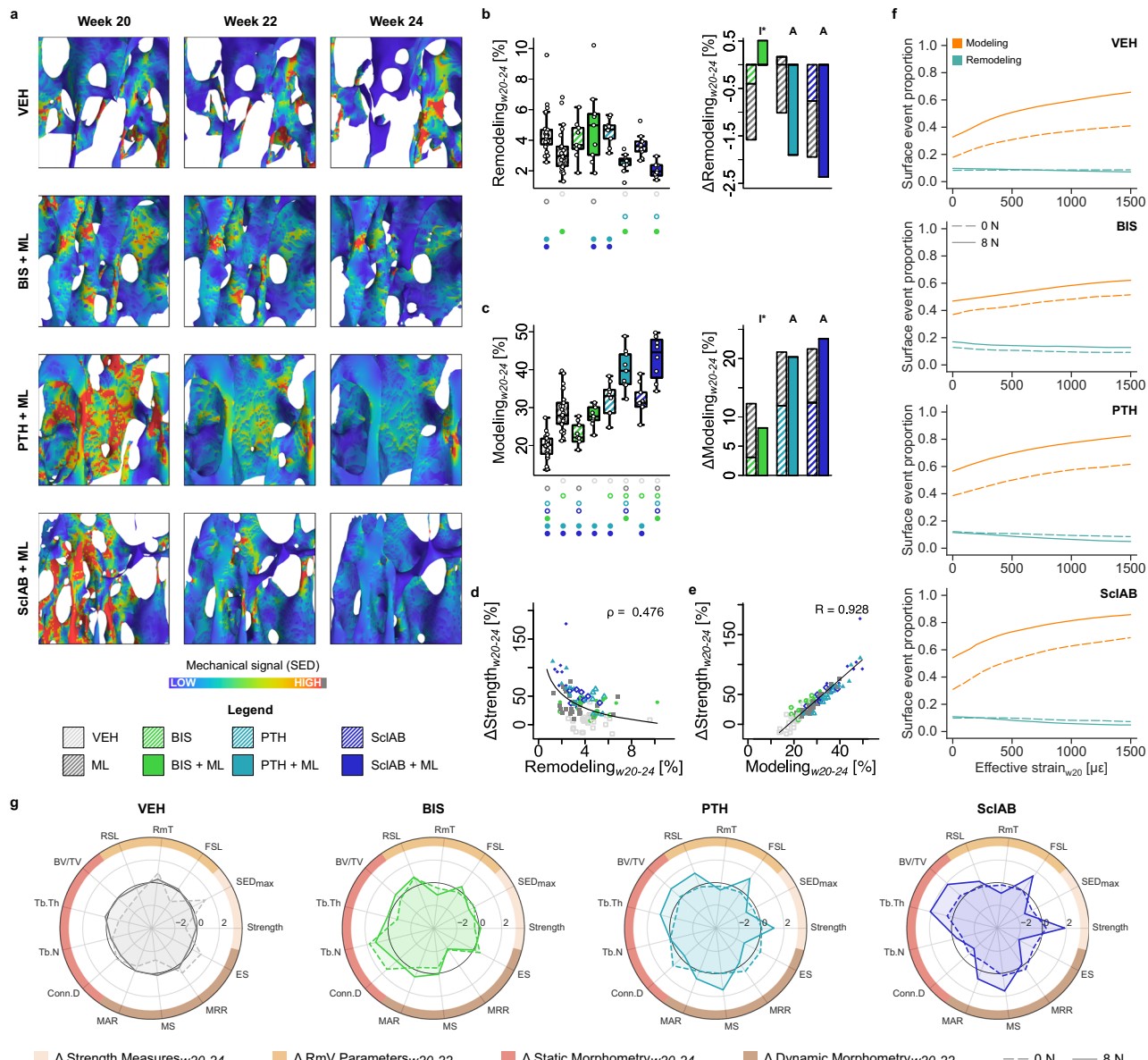

**Fig. 6 | Modeling capacity correlates highly with the increase in predicted strength. a** Heterogeneity of the strain distribution decreases to different extents in different treatments. Qualitative comparison of changing SED-distribution for VEH, BIS + ML, PTH + ML and SclAB + ML over the course of 4 weeks of treatment. **b** Remodeling as calculated from formation following resorption at the same location over three measurement time points (w20, w22, w24). **c** Modeling as calculated from formation following quiescence at the same location over three measurement time points (w20, w22, w24) (b-c) Each data point represents one animal (biological replicate; VEH: $n = 25$, ML: $n = 26$, BIS: $n = 9$, BIS + ML: $n = 9$, PTH: $n = 10$, PTH + ML: $n = 9$, SclAB: $n = 9$, SclAB + ML: $n = 8$). The box spans the interquartile range (IQR) from the 25th to the 75th percentile. The median is shown as a line inside the box. Whiskers extend to the most extreme data points within 1.5 ×

IQR, points beyond this range are plotted as outliers. Δ denotes change from VEH; sum of single vs. combined treatment group means indicating graphically antagonistic (I), additive (A) or synergistic (S) effects. Significant differences are depicted with a symbol in the color of the corresponding group (o: to monotreatment, ●: to combined treatment, see Supplementary Dataset S.1 for color encoding, $p < 0.05$). **d** Remodeling amount in % correlated to increases in predicted strength fitted using a power-law function. **e** Modeling amount in % correlated to increases in predicted strength fitted using linear regression. **f** Surface event proportion (modeling or remodeling) in relation to the mechanical signal. **g** Spider plots summarizing the mean of outcome parameters of the analyzed treatments. Values were standardized to facilitate visual comparisons between groups. Source data are provided as a Source Data file.

## Discussion

The main goal of osteoporosis therapy is to restore bone mass to reduce fracture risk. To achieve this efficiently, existing bone must be preserved and new bone formed at mechanically relevant sites. In healthy bone, bone (re)modeling is proposed to be regulated via the mechanostat[6,19–21]. Here, we investigated how three current pharmacological treatments influence this regulation in the initial treatment phases in osteoporotic mice, particularly when combined with mechanical loading. Our findings suggest that all three investigated

treatments induce bone formation and inhibit bone resorption at the mechanically relevant sites, but it was only in anabolic therapies that mechanical loading additively (PTH) or synergistically (SclAB) enhanced predicted bone strength.

When combined with sclerostin antibody treatment, mechanical loading led to a synergistic increase in predicted bone strength and an additive increase in bone volume fraction. Increases in mineral apposition rates and reductions in eroded surfaces were synergistic, consistent with previous findings[12,22,23]. This response may be explained by

dual suppression of the activity of sclerostin, a protein secreted by osteocytes that antagonizes Wnt signaling[11]. While mechanical loading independently suppresses sclerostin expression in osteocytes[13], SclAB pharmacologically neutralizes any residual sclerostin[11]. Together, these effects enhance Wnt/β-catenin activity and likely contribute to the pronounced anabolic response[24]. Additional pathways may also be involved, as suggested by studies in SOST-KO mice, which still showed loading-induced bone formation[25]. One candidate is Dickkopf-1 (Dkk1), another Wnt inhibitor[26–28]. Robling et al.[13] reported that mechanical loading reduced both *Sost* mRNA (-73%) and *Dkk1* (-49%) transcripts. Since both proteins compete at LRP5/6 receptors in osteoblasts and osteocytes, their simultaneous inhibition may underlie the observed synergy between SclAB and mechanical loading.

We found additive increases in predicted strength and increased sensitivity of mechanoregulation mechanisms for PTH + ML, which is in line with previous results where PTH increased the mechanoresponse of osteoblasts only when combined with fluid flow[29]. Hagino et al[30] hypothesized that PTH lowers the (re)modeling threshold of the (re)modeling velocity function, which our findings support (Fig. 5). In previous animal studies, mechanical loading combined with intermittent PTH (1-34) treatment led to synergistic effects in the trabecular compartment of the healthy tibia, but it was only prevalent in regions that were not remodeled, but modeled[31].

Bisphosphonates were the only treatment that preserved Tb.N but showed no additive effects with mechanical loading, consistent with clinical studies reporting limited benefit of combining bisphosphonates and exercise[32,33]. Since the mineralizing surface was significantly increased in the bisphosphonate group compared to the vehicle group, our results suggest that bisphosphonates not only affect osteoclast activity but also osteoblast activity, possibly through disruption of the coupling process via elevated osteoclast apoptosis[34], potentially impairing the spatial guidance of osteoblasts by the mechanical signal. The observation that the BIS and BIS + ML groups exhibit RmT values as low as those of other treatment groups, despite showing the least overall (re)modeling response, may indicate a compensatory increase in osteocytic mechanosensitivity, potentially allowing selective resorption of mechanically less relevant areas.

This study focuses on osteoblast and osteoclast surface activities using in vivo micro-CT, which allows longitudinal tracking of bone (re)modeling. Histological analysis could offer further insight into cellular and tissue-level mechanisms to support the cell-specific interpretations derived from the micro-CT data. However, due to the tissue preservation method and the limited number of available samples, it was not considered feasible in the current study. It's worth noting though that previous work demonstrated a significant correlation between micro-CT-derived and histomorphometry-based MS and MAR (R ≈ 0.59 – 0.78)[35,36]. Furthermore, Birkhold et al.[37] have demonstrated the ability of longitudinal in vivo micro-CT to reliably quantify bone (re)modeling, i.e., separation into modeling and remodeling, in both trabecular and cortical bone. While in vivo micro-CT does not allow direct visualization of osteocytes, our measurements reflect the functional output of the bone cells in which osteocytes play an important role through orchestration of bone formation and resorption in response to local mechanical stimuli[6].

Regions with high local strain are linked to higher occurrences of microscopic tissue damage[38] of which even small amounts can lead to large reductions in strength[39] and increase fracture risk[40]. Reductions in SED over time do not necessarily imply a decrease in bone formation capacity but rather reflect system saturation, consistent with mechanostat theory, where bone formation plateaus once strain thresholds are reached.

As a limitation, our developed system does not yet incorporate the level of the transcriptome[41]. Other cell types, particularly immune cells and endothelial cells, also contribute to bone (re)modeling via inflammatory signaling and vascularization, respectively. Recent work originating from fracture healing highlights the role of immune and vascular interactions, particularly in the context of aging and mechanical loading[42]. While these cell types were not analyzed here, future multimodal approaches could complement imaging-based assessments.

Given the biological differences among the therapies selected, including administration routes and duration limitations, direct long-term comparisons are not the aim of this study. Instead, our focus on short-term effects aimed to investigate potential synergistic improvements in bone strength and to gain mechanistic insight into how pharmacological and mechanical stimuli modulate mechanoregulation. The findings support the mechanostat theory: treatments lower mechanical setpoints, enabling bone formation at lower strains. However, only anabolic treatments effectively reinforce highly loaded trabeculae and reduce peak strains, underscoring the importance of modeling capacity for strength gains. And while our results highlight the promise of mechanical loading as an adjunct to anabolic therapies, clinical studies will ultimately be required to determine optimal exercise modalities, just as pharmacological dosing requires careful optimization.

In conclusion, the current study revealed that drug treatments target trabecular bone (re)modeling at mechano-functionally relevant sites, and osteoanabolic treatments, in combination with mechanical loading, additively and synergistically increase the bone response. The additive (PTH) and synergistic (SclAB) increases in predicted strength can be explained by reductions in resorption rates, shifts in the (re)modeling thresholds as described by Frost in the mechanostat theory more than 30 years ago, and by the modeling ability of anabolic treatments. Linking organ-level changes induced by physical or pharmacological therapies to the mechanosensitivity of cells provides insight into how well skeletal structures can adapt to mechanical demands under pathological or treatment conditions, a task for which the body has optimized itself over millions of years of evolution. Load-bearing exercise may be a promising adjunct for physically active patients with osteoporosis on osteoanabolic therapy.

## Methods
### Study design
The study design is illustrated in Fig. 1a. In total, 120 female 15-week-old C57BL/6 mice (n = 60, C57BL/6JRccHsd mice from RCC/Harlan, Füllinsdorf, Switzerland/Horst, The Netherlands; n = 60, C57BL/6JRj mice from Janvier Labs, Le Genest-Saint-Isle, France) were subjected to bilateral ovariectomy. A basal micro-CT image of the 6th caudal vertebra was obtained (vivaCT 40, Scanco Medical, Brüttisellen, Switzerland). Recovery lasted five weeks after surgery, as in this period, an osteopenic state was established in the trabecular compartment of the 6th caudal vertebra in a previous study[43]. Pinning of the adjacent vertebrae as preparation for the loading procedure took place in week 17. At week 20, the animals were divided into 8 groups and the following treatments were administered for the following four weeks: BIS (n = 9), BIS + ML (n = 9), PTH (n = 10), PTH + ML (n = 9), SclAB (n = 9), SclAB + ML (n = 8), ML (n = 26), VEH (n = 25). Ten mice were excluded due to swollen tails or loose pins. In the SclAB study, four mice showed a periosteal reaction, possibly due to injury of the periosteum during intravenous injection at the tail vein. One mouse in the BIS + ML group did not lose bone volume after ovariectomy. These mice were excluded from evaluation, resulting in 105 animals included. Micro-CT was performed in weeks 15, 20, 22 and 24. Surgeries, imaging, and loading were performed under isoflurane anesthesia (induction 4%, maintenance 1.5-2.5%, Attane, Piramal Healthcare, Mumbai, India). Vehicle control was an isotonic buffer (pH 5.3). While the data were obtained across four experiments (at different times), the study and loading parameters were not different, except for the administered pharmacological treatments, which were combined in this study. The bone responses of the VEH and the ML groups of both mouse lines (JRccHsd and 6JRj) showed no differences in reactions (ΔBV/TV$_{w20-24}$; VEH:

$p = 0.937$; ML: $p = 0.206$; two-way ANOVA followed by Tukey's post-hoc test), thus ML groups from all experiments and VEH groups from all experiments were merged, explaining the larger sample size in these groups. The merging was conducted because separate vehicle group presentations would not have yielded additional insights but substantially would have increased data complexity (see Supplementary Dataset S.3). All animal experiments were performed with approval of the local authority (Kantonales Veterinäramt Zürich, Zurich, Switzerland, license number 171/2008, 108/2011, 078/2012) and reported according to ARRIVE 2.0 guidelines.

## Animal care and monitoring

Animals were kept in groups of 3–5 mice in filter top cages in Scantainers (Scanbur, Karslunde, Denmark) with *ad libitum* access to tap water and chow (KLIBA NAFAG 3437, Granivit AG, Kaiseraugst, Switzerland), wood chips bedding and a 12 h:12 h light:dark cycle. Room temperature was 22 °C ± 2 °C and humidity was not controlled. Cages were changed weekly by the experimenters. Microbiological status of the facility (Animal Imaging Center, ETH Zurich) was not monitored, because it housed only animals during experiments. Animals were checked and scored daily for three days after OVX surgery and pinning and at least three times per week during the further experiment. Meloxicam 2 mg kg⁻¹ was injected s.c. for three days after OVX and pinning.

## Mechanical loading

A validated in vivo mouse tail loading model was used to induce trabecular bone adaptation under controlled mechanical conditions[43–45]. Stainless steel pins were implanted into adjacent vertebrae under fluoroscopic guidance three weeks prior to loading. An 8 N cyclic axial load (3000 cycles, 10 Hz) was applied to the 6th caudal vertebra (CV6) using a custom loading device (caudal vertebra axial loading device, CVAD[44]) three times a week. Control animals (0 N groups) underwent identical procedures, including pin placement and anesthesia, but no load was applied. A force of 8 N was used in order to maximize the effect within a short timeframe. It was the maximum force that could be applied based on what the pins within the bone could bear, and represents twice the force exerted on mouse vertebrae under physiological conditions[46]. Pistoia simulations estimate a maximum failure load of 24.5 N at CV6 in osteoporotic mice at week 20 (predicted trabecular strength, see Supplementary Dataset S.2).

## Pharmacological treatment

The BIS group received 100 µg kg⁻¹ zoledronate (Zometa 4 mg/5 ml; Novartis Pharma Schweiz AG, Rotkreuz, Switzerland) subcutaneously once in week 20. Corresponding VEH (0.9% NaCl, B. Braun Medical AG, Sempach, Switzerland) was injected at the same time point. Mice in the PTH study received 40 µg kg⁻¹ PTH (hPTH 1-34, Bachem AG, Bubendorf, Switzerland) or the according VEH (0.9% NaCl) subcutaneously daily over four weeks. SclAB (100 mg kg⁻¹, Novartis Pharma AG, Basel, Switzerland) or the corresponding VEH (isotonic buffer, pH 5.3) was injected i.v. weekly into the lateral tail vein over 4 weeks.

BIS and SclAB doses were selected based on established protocols in the literature[24,47]. The PTH dose was determined in a dose-response study comparing 10 µg kg⁻¹ d⁻¹ and 40 µg kg⁻¹ d⁻¹, following an initial 80 µg kg⁻¹ d⁻¹ dose proposed in the literature[48] that led to excessive increases in bone volume fraction (BV/TV) by the end of the experiment. The 40 µg kg⁻¹ d⁻¹ dose was chosen to obtain BV/TV increases comparable to those observed with BIS and SclAb treatments, enabling a more balanced comparison across treatment groups (see Fig. 1b).

## Bone morphometry

Micro-CT images were obtained with a vivaCT 40 (Scanco Medical AG, Brüttisellen, Switzerland) operated at 55 kVp and 146 µA. 1000 projection images were obtained with an integration time of 200 ms.

Grayscale images were reconstructed using a Feldkamp algorithm with automated beam-hardening correction, resulting in an isotropic voxel size of 10.5 µm. The image of week 20 was aligned in the main loading direction. Follow up scans were registered to the previous scan. A constrained 3D Gauss filter (sigma 1.2, support 1) and a global thresholding procedure of 22% (approximately 580 mg HA cm⁻³) were applied. Automated masks were created to extract the trabecular bone volumes (trabecular mask truncated by 10% of length at both ends to exclude epiphyses and primary spongiosa; middle 26% removed because this area is almost void of trabeculae)[18]. The cortical mask included 75% of the length of the inner mask[18]. Static trabecular bone morphometric parameters were calculated according to Bouxsein et al.[49]. Dynamic bone morphometry was performed by registering and superimposing segmented bone volume images of consecutive time points. Bone formation and resorption rates were calculated[36] over two two-week periods (w20-22 and w22-24).

## Finite element analysis of mechanical properties

Finite-element simulations were used to analyze the mechanical properties of the vertebrae. 3D micro-FE models were generated by converting all voxels of the micro-CT image to 8-node hexahedral elements, with each model consisting of approximately 1.8 million elements. A Young's modulus of 14.8 GPa and a Poisson's ratio of 0.3[44] were assigned. Each model was solved with ParOSol[50] running on the Euler cluster operated by Scientific IT Services at ETH Zurich. Cylindric intervertebral disks (10% of absolute height) were applied and assigned the same Young's modulus and Poisson's ratio as bone. The top of the model was displaced by 1% of the total vertebral length, while the bottom was fixed. The resulting reaction force was rescaled to the value of the force applied in the experimental setup; consequently, all the FE outcomes were rescaled accordingly (4 N for non-loaded[46] and 8 N for loaded mice). Strain energy density (SED), defined as the increase in energy associated with the tissue deformation per unit volume, and effective strain were used as the mechanical signal. Trabecular bone strength was estimated for each mouse at each time point by using the Pistoia criterion[51], which assumes that bone failure is initiated if a certain percentage of the bone tissue (2%) is strained beyond the tissue yield effective strain (7000 µε). This criterion has been shown applicable to the mouse vertebra as well[52]. For the predicted bone strength estimation, the full bone was modeled but only the effective strains in the trabecular compartment were analyzed.

## Local mechanoregulation analysis

The local mechanical environment was derived from micro-FE simulations of the baseline scan. Here, SED is used as a mathematical term to describe the (re)modeling stimulus phenomenologically. To quantify the relationship between the local SED and cellular activity on the bone surface, the bone formation or resorption sites determined by rigid registration were projected onto the surface of the baseline scan, resulting in three masks representing three different clusters of formed (F), quiescent (Q) or resorbed (R) bone surfaces[6]. In short, the mean SED value was calculated in each of these three masks and for each mouse, and normalized to the mean SED of the quiescent surface as absolute SED values differ per mouse and over time due to differences in BV/TV. To establish a quantitative description of the mechanoregulatory system, the relation between increasing mechanical stimuli and consequent (re)modeling events was assessed by calculating the relative percentage of voxels being formed, quiescent and resorbed for each value of SED binned at 1% step size of the maximum SED, and assuming that each (re)modeling event has the same occurrence probability (i.e., formation, resorption and quiescent regions were virtually rescaled to have the same amount of voxels) to remove dependencies on imbalances in the (re)modeling process due to differences between animals or treatments. The conditional probabilities were fitted per animal by exponential functions. Correct classification

rate (CCR) values were calculated from the mean conditional probabilities from all animals, returning a scalar value between 0.33 and 1 assessing overall mechanoregulation, with 0.33 meaning (re)modeling is independent of the mechanical signal and 1 meaning it is entirely dependent on the mechanical signal. (Re)modeling velocities (RmVs) and their mechanostat parameters were calculated according to Marques et al. for each animal[19]. Minimum and maximum RmV, represented as formation saturation level (FSL) and resorption saturation level (RSL), as well as (re)modeling thresholds (RmTs) were extracted from the resulting curves.

## Statistical methods
Statistical analysis was performed using R (R, Auckland, New Zealand[53]). Data are presented as group means ± standard error unless otherwise specified. P-values lower than 0.05 are considered significant; statistical significance was defined as $*p < 0.05$, $**p < 0.01$, $***p < 0.001$. Normality was assessed using the Shapiro-Wilk test (in R, with $\alpha = 0.05$).

Static morphometric and mechanical parameters of trabecular bone in the 6th caudal vertebra were measured at weeks 15, 20, 22, and 24, with $\Delta$ representing the percentage change between weeks 20 and 24. Two-way ANOVA followed by Tukey's post-hoc test was used to compare $\Delta$w20-24. Synergy (S) or antagonism (I) between treatment and mechanical loading was evaluated using the interaction term ("treatment:loading") in a linear model.

A synergistic effect was defined as a combined treatment response that exceeded the sum of the individual effects of treatment and mechanical loading. This implies that the two interventions may act through the same or facilitating biological pathways. Statistically, such synergy is indicated by a significant and positive interaction term between the main effect (treatment) and mechanical loading. An antagonistic effect was defined as a combined treatment response that was less than the expected additive effect of treatment and mechanical loading. Statistically, such antagonism was reflected by a significant and negative interaction term between the main effect (treatment) and mechanical loading. The animal provider from the third (SclAB) and fourth (PTH) experiment was different; therefore, we adjusted the model for this.

Dynamic morphometric parameters were analyzed during weeks w20–22 and w22–24. One-way ANOVA followed by Tukey's post-hoc test was applied to absolute values in each time interval. Synergy or antagonism was assessed using the interaction term from a linear model of $\log_2$-transformed values in each time interval.

SED in formation (F), quiescence (Q), and resorption (R) surfaces were reported for each group in each time interval. Differences between F, Q, and R were compared using two-sided Wilcoxon signed-rank tests with Bonferroni correction after testing for normality.

For conditional probabilities, exponential curves were fitted to the formation and resorption slopes. For formation, a saturating exponential model $f(x) = y_0 + a \cdot (1 - \exp(-bx))$ was applied to the first 40% of normalized SED data for each animal individually, with parameters $y_0$, $a$, and $b$ representing level of formation in the absence of mechanical stimulus (i.e., at SED = 0), maximal formation, and sensitivity to mechanical stimulus, respectively. For resorption, a decaying exponential model $f(x) = y_0 + a \cdot \exp(-bx)$ was used to assess the rate of resorption. Resorption asymptote $\min_R$ represents the saturation level of resorption at high mechanical signal, i.e., the minimum resorption probability ($x \rightarrow \infty$), indicative of the amount of non-targeted resorption. Coefficient $a$ represents the magnitude of the response, i.e., the difference between the initial and the minimum resorption probability. Coefficient $b$ is the rate of decay, representing how quickly the probability of resorption drops as SED increases. Synergy or antagonism was assessed using the interaction term from a linear model applied to $\log_2$-transformed coefficients, with significance determined by one-way ANOVA followed by Tukey's post-hoc test.

Formation and resorption saturation levels (FSL, RSL) and remodeling thresholds (RmTs) were derived from hyperbola remodeling velocity curves, fitted with the function $f(x) = y_0 + \frac{a}{(b-x)}$, where $x$ represents the mechanical signal, $y_0$ the FSL, and $a$ the (re)modeling velocity modulus. The remodeling threshold (RmT) is defined as: $RmT = b + \frac{a}{y_0}$. One-way ANOVA followed by Tukey's post-hoc test was used for statistical comparisons, with synergy or antagonism assessed using the interaction term from a linear model applied to absolute values in each time interval, because FSL and RSL can obtain negative values.

Modeling and remodeling amounts were calculated by monitoring spatial agreement of voxels over the time intervals w20-22 and w22-24, following the approach presented by Birkhold et al.[37]. Modeling amount was defined as formation without prior resorption in either the first or the second interval, divided by the amount of surface voxels. The remodeling amount was defined as resorption followed by formation, divided by the number of surface voxels.

Linear correlations between the %-change in predicted strength and parameter outcomes were assessed using Pearson's correlation coefficient R. For nonlinear relationships, we employed a power-law model of the form $y = a \cdot x^b$, which was linearized via logarithmic transformation to $\ln(y) = \ln(a) + b \cdot \ln(x)$. Parameters $a$ and $b$ were estimated using linear regression on the transformed data, and the Spearman's correlation coefficient $\rho$ between the log-transformed observed and predicted values was used to assess the goodness of fit. Since the fitted power-law model adequately described the nonlinear relationships, we did not perform a formal model selection procedure to compare alternative models, such as those based on the Akaike information criterion (AIC).

## Data availability
The primary data supporting the findings of this study are available within the paper. The raw data generated during the study are available in a Zenodo repository (see link below). Source data are provided with this paper.

## Code availability
All data and code are available in a Zenodo repository: https://doi.org/10.5281/zenodo.17256047.

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

## Acknowledgements

We acknowledge Kathleen Koch for assistance and performance of animal experiments of the PTH and BIS groups, and Dr. Ulrike Kettenberger for assistance and performance of animal experiments of the PTH group. Nirujan Pasupathy and Júlia van den Nest Molina have supported the analyses of modeling and mechanoregulation mechanisms. This work was funded by the European Union grant VPHOP (FP7-ICT-2008-223865; FAS, FML, CW, GAK) and the European Research Council ERC Advanced grant (MechAGE ERC-2016-ADG-741883; FCM, GAK).

## Author contributions

RM, GAK, and PJR designed the experiment. CW, GAK, MS, and FML performed the experiments. FAS, FCM, JG, CW, MS, FML, and GAK performed image processing and analysis. MK provided pharmacological agents. RM, GAK, PJR, and CK supervised the project. FAS wrote the original draft of the manuscript. All authors contributed to data interpretation, were involved in discussions throughout the study, and participated in manuscript revision and preparation.

## Funding

## Competing interests

MK, PJR, and CK are employees of Novartis. The other authors declare they have no competing interests.
