## [Transparent Peer Review file · Nature Communications]

Combined physical and pharmacological anabolic osteoporosis therapies increase bone response and mechanoregulation in female mice

Corresponding Author: Professor Ralph Müller

Version 0:

Reviewer comments:

Reviewer #1

(Remarks to the Author)

The manuscript by Schulte et al presents a novel comparison of different osteoporosis treatments in ovariectomized mice and put them in addition under mechanical compression loading. The aim of the work is to compare anabolic vs. catabolic osteoporosis treatments with or without loading (as surrogate to training intervention). Compared were bisphosphonate (BIS), parathyroid hormone (PTH), sclerostin antibody (SclAB) or vehicle (VEH) treatment in ovariectomized 20-week-old C57BL/6 mice.

In clinical routine treatment of osteoporosis patients and doctors are frequently unsure if an additional physiotherapy or training would help to maintain bone mass and thus a very practical reasoning is behind the here presented study. The work is organized in a sound manner and the question targeted has very practical consequences.

While I do like the approach I do have two main concerns, that could eventually be addressed by the authors:

- Why did you use the selected therapies in this specific comparison? Did you formulate initially more concrete hypothesis for each individual comparison loaded vs. sham in bisphosphonate (BIS), parathyroid hormone (PTH) and sclerostin antibody (SclAB)? E.g. regarding OB or OC or synergistically? A more stringent biological explanation for the comparisons would be helpful
- Some treatments come with side effects; not all can be given for a similar (long) period. And not in a similar manner. This is due to substantial biological differences across the therapies selected. This aspect seems to be widely ignored; eventually focusing here on more comparable approaches and eventually only the initial effects (and not long-term consequences) might help?
- If the work focuses only on osteoblasts and osteoclasts and their activities, I would assume that more in depth analyses on the tissue (histological) level could help confirm the findings from the μ CT data.
- In my eyes very interesting findings are – specifically in the light of the authors reference to Frost – the separation that formation – resorption volumes see in the PTH + ML and SclAB + ML (Figure 4), while they do overlap in almost all other cases. So maybe Frost was correct in the end – but only in a co-stimulation with PTH or SclAB. What sets these stimuli in respect to mechano-responsiveness apart from the others?
- The manuscript would benefit from a clear link to biology, here specifically osteoblastic or osteoclastic activities. In one sentence, the authors mention osteocytes. But there is no reasoning given on how specifically Osteoblasts or osteoclasts are affected. Frost referred more to the osteocytes, which are here in analyses and data not at all in the focus. I am a bit surprised that the biology is not really included, even though the therapies analyzed are very distinctly different in their effects on these cell types.
- Finally, no other cells – such as immune or endothelial cells– are here considered. This limitation needs more discussion.

(Remarks on code availability)

no further comments

Reviewer #2

(Remarks to the Author)

This work from Schulte et al examines combined effects of experimental mechanical loading with three current therapeutic approaches for osteoporosis in an animal model. This is an important question to test, and the experimental intervention they have carried out is good. The authors report that the combination of two anabolic therapies (anti-sclerostin and intermittent PTH) with mechanical loading has a (numerically) more than additive effect on trabecular bone volume and on cortical thickness. They then carry out a secondary analysis which shows that bone formation occurs where there is high predicted mechanical signal, and bone loss where there is less mechanical signal; this is confirmatory data of a general principle of bone mechanobiology, but they take it one step further in attempting to develop a data-driven theoretical model of whether there is a change in the set-point at which bone surface activity occurs. This latter part of the paper is extremely difficult to critique as it is not very clearly described. The experiments are well planned, and the discussion section of the work is very thorough, but the data presentation is unclear and I disagree strongly with the way the authors have interpreted the combined effects of the treatments they have tested.

Major concerns:

1. Claims of synergy and interference: The work lays claim to a synergistic effect of mechanical loading and anabolic therapy. This is an overclaim and shows a lack of understanding of synergy. Usually, in pharmacological analysis, proof of synergy depends on showing that the effect of two treatments together is much greater than the addition of the two therapies, and is based on the idea that they work through the same pathway. In this project, they have calculated the average increase of the combined treatment, and compared it numerically to adding together the average effect of each treatment. There is no statistical analysis to determine whether these "effects" are significantly different. What is also a problem is that they use this method on parameters, such as trabecular number and trabecular thickness, which are not separable, and each contribute to trabecular bone density. This overlooks the fact that an increase in trabecular thickness is frequently associated with a reduction in trabecular number, since it can reflect a loss of thinner trabeculae, rather than a specific increase in trabecular thickness. It is not possible to test for synergy on such parameters in this context. The meaning of "interference" is not clear.
2. Data presentation: The presentation of much of the data as percentage change from baseline obscures the observations being made here; the data is not at all transparent, though I appreciate that an in vivo study allows you to see changes over time. Inclusion of the raw data in Supplementary Figures is needed for data transparency.
3. It seems that three different vehicle groups have been combined into a single vehicle group of samples. This is not good practice, and will have a major effect on the statistics for this study. The vehicle groups should be shown separately, at least to confirm that there was no difference between the three different vehicle groups.
4. It is surprising that, in the single agent treatment groups, that there was a reduction in BRR/BV and ES with PTH treatment, and a reduction in MAR with anti-sclerostin. This needs to be discussed.
5. The paper is not written for a general audience, apart from the introduction. The results section is highly technical and full of jargon, making it extremely challenging to follow. What is the "natural law" that you refer to in line 250, for example? I found it almost impossible to understand the final two figures, and their description on the results section, despite being very interested in the outcomes. Part of this is the continuing mis-use of the term "synergy", but also these sorts of highly theoretical methods need to have some clear explanation for non-theorists, particularly to justify the mathematical approach being used.

Minor points

1. The authors make many comments in the introduction and discussion which are not supported by references (e.g line 58, 78, 82...)
2. The introduction to this paper reads more like an opinion piece rather than a critical review of the literature. This needs attention.
3. Some references are not appropriate - for example, the peculiar term (re)modeling (which is common, but confusing) was not used in reference 9. Also the references used in line 93 do not really support the statement being made here.
4. The use of the term (re)modeling is not really helpful in this manuscript - it is not clear whether you are talking about modeling or remodeling or simply bone surface activity. If you cannot distinguish between the two with this method, I think it is better to say "bone surface activity".
5. Use of the term "pharmaceutical" is inaccurate - it refers to pharmacological agents used in humans. The correct term here would be "pharmacological".
6. The inclusion of data in both the text and the figures is very confusing. It is better to simply report the data, and allow the reader to look at the figures.
7. How were the doses of bisphosphonate, PTH and Anti-sclerostin selected?

(Remarks on code availability)

I did not see any supporting code.

Reviewer #3

(Remarks to the Author)

Overview of Key Results and Significance

The aim of the manuscript was to evaluate the interaction between mechanical loading and osteoporosis drug treatments in the vertebral trabecular bone of osteoporotic female mice, and test whether mechanical loading acts additively or synergistically to improve drug treatment outcomes. The authors have developed, over several years, innovative biomechanical, imaging and statistical modeling approaches that allow them to address this topic in a detailed and rigorous manner. They have selected three relevant osteoporosis drugs (two anabolic, one anti-catabolic), alone and in combination with mechanical loading, which enhances the generalizability of the findings. The main finding is that combining mechanical

loading with anabolic osteoporosis drug therapy synergistically enhances bone volume and bone (re)modeling, whereas no such effects were observed with anti-catabolic drug. In addition to addressing this important topic (ie, effects of drugs + loading), the authors have framed the work in the context of the influential “mechanostat model” proposed by Harold Frost. Frost based the model largely on observations and inference (and little data). Herein, the authors have used their valuable data set to provide a rigorous test of the model, and show that its basic tenets align well with data and are indeed a useful framework for explaining (and predicting) how bone responds to loading, and how osteoporosis drugs shift this response. Further, the data provided by this manuscript is an important addition to the literature as it addresses the clinically relevant question about the role of physical therapy in osteoporotic skeletons undergoing pharmaceutical treatments. There are issues with the clarity and presentation of results, figures, etc., which make parts of the paper challenging to follow. Most of the comments below highlight these issues.

Major Comments:

1. The results section can be difficult to follow, especially with the frequent inclusion of mean \pm -SD values for all the groups (which is redundant with same data shown in Figures). Consider stating the % difference between groups, rather than listing the mean/SD values.
2. The results include many statements of difference where it is unclear if the differences are supported by statistical testing. Please use the word ‘significant(ly)’ to clarify (e.g., lines 120-122).
3. What is the rationale for limiting the SED/SEDmax range to 0.0-0.4 (eg, Fig 5)?
4. The in vivo micro-CT was done at 20, 22 and 24 wks age. It is not always clear which time interval is used for calculation of dynamic indices – please add this information as a label on the relevant figures. (It is stated in the captions but is hard to find.) If different intervals are used in different sections of the Results, state the rationale and if the results are sensitive to the interval used. This is an important concern that must be addressed. For example, in Fig 1 caption it indicates w22-w24 was used to determine remodeling rates in panels g-l. Figure 3 does not mention the interval for panels a-f, and in Figure 4, it indicates w20-w22 was the interval used to determine formed/quiescent/resorbed bone. Please justify the use of interval used, and state clearly in the figure captions.
5. Addition of supplemental data would enhance the rigor of the paper.

Specific Comments:

Introduction

6. The Introduction could benefit from additional details about the PTH analog and bisphosphonate, similar to the level of detail provided for the sclerostin-targeting drug.
7. The section of the Introduction from lines 60-97 could be shortened. In general, the Introduction is too long.
8. Lines 53-55: State here that sclerostin is a Wnt antagonist. Wnt binds to the co-receptors LRP5 (or LRP6) and Frizzled.
9. Lines 75-78: The mention of microdamage associated with bisphosphonates requires supporting citations. This is an important issue, albeit with limited clinical data.

Results

10. ‘Dependence of (re)modeling probability on mechanical stimulus’ and ‘Synergisms in (re)modeling velocity and mechanostat parameters’ sections could benefit from more descriptive headers and clear summaries or key takeaways at the end of each section to help guide the reader through the findings.
11. In ‘Dependence of (re)modeling probability on mechanical stimulus’ section, discussion about Fig. 5e, CCR is missing. Additionally, the implication of changes in non-targeted formation and resorption (Fig. 5i and j) should be discussed in the result. Many readers will not understand what you mean by “non-targeted formation/resorption”.
12. Lines 127-131 and Fig 1d,e: The decreases in Tb.N and Conn.D for most groups are surprising to me. Please add some explanation or context. (Is the increase in BV/TV due solely to increased Tb.Th, and despite fewer trabeculae?)
13. Figure 1b-e. The y-axis values do not make sense. Should 100% be 0%, ie., representing change from week 20 (start of treatment)?
14. Fig 1b-e. In addition to between group comparisons noted by ‘o’, consider adding a symbol indicating whether the delta for each group is different from zero (i.e., the treatment increase (or decrease) the parameter).
15. Lines 139-145: Please state how Eroded Surface is defined.
16. Section starting with line 162 and Figure 2: The comparisons of combination groups vs. the sum of the monotherapies to infer Interfering, Additive or Synergistic effects is an excellent concept, but it is unclear how this is implemented and additional justification should be stated in the Results. Is the height of each bar the mean value for that group? It is unclear if these comparisons can be supported with statistical testing to increase rigor. Also state in Figure what the time interval used is for the data in the bar graphs.

17. Lines 184- : The measures of trabecular “strength” estimated/predicted using FEA, and should be referred to as ‘Predicted strength’.
18. Line 189: Suggest that you add a statement explaining what is meant by an ‘interfering’ effect, and how you interpret this.
19. Figure 2j not referenced in the text.
20. Line 214: Lowered compared to what?
21. Line 215: Is “beneficial” correct? Graph looks like Interfering.
22. Line 224: Typo “inlets” “insets”.
23. Fig 3h: the second graph shows ‘stderr’ rather than R^2 .
24. Line 240: Word “ranges” is vague here.
25. Lines 250, 252: The term “natural law” is an overstatement. Please find another way to state this.
26. Figure 4: It is unclear what %diff SED is showing; it is explained in the text, but could be also stated in the figure caption. It looks like the average value of %diff SED for quiescent group is not always equal to 0%. If so, can you explain how this can be.
27. Section starting on Line 260 Dependence of (re)modeling probability on mechanical stimulus. This is one of the most interesting aspects of the study, but the presentation is challenging as it lacks quantitative values (Figs 5a-d) and is unclear if statistical testing can be performed to support the comparison of AUC values and probabilities of non-targeted formation and values of RQ and QF.
28. Figure 5a-d: Include equations of these exponential curve fits in Supplemental. Do constants of these eqns have meaning that can support the qualitative descriptions of the curves in the text (e.g. Line 270-271). In general, the qualitative descriptions and the statements referring to steeper slopes, etc, lacks quantitative rigor. Authors should mention the slope values to give the readers the opportunity to better appreciate the magnitude of differences in mechanical sensitivity among the groups.
29. Figure 5e,f,g are not cited in the text.
30. Lines 276-284: If you want to include the AUC values, it would be better in a table than in the text.
31. Line 285: Suggest revise to state “In ML, the probabilities of non-targeted formation...”
32. Fig 6f-h. Unclear that these RmV curves are relative to the VEH curve. State in the caption.
33. Figure 6 and related text (lines 349-357): Values of RmV seem very low (eg, 0.00585 $\mu\text{m}/\text{day}$). Are these correct? Why so different from histological mineral apposition rates which are on order of 1 $\mu\text{m}/\text{day}$?
34. The implication of changes in RSL and FSL (Fig. 6i and j) should be discussed.

Discussion

35. Although the BIS and BIS+ML groups exhibit the lowest RmT values, suggesting a lower threshold for remodeling, they also demonstrated the least overall response. This apparent discrepancy should be addressed in the discussion section.
36. Fig 5a-d: Since the differences in conditional probabilities between the monotreatment and the loading + treatment groups are relatively small (approximately 1–2%), the authors should consider commenting on the biological or practical significance of these differences in the discussion section.
37. Line 370-71: “this is regulated via the mechanostat” – can you support this with citations, or qualify it “proposed to be regulated via the mechanostat”. This study is the most convincing I have seen to support that this is true.
38. Line 402: “46-bladed propellers” seems typo. Usually described as “4-beta propeller structures”
39. The SED maps in Fig. 5m show a reduction in SED over time. Given that the manuscript argues higher SED is associated with increased bone formation, does this observed reduction imply a decrease in bone formation capacity over time?

Methods

40. Lines 552-554: The method for bone strength estimation seems very sound, but is based on assumptions about failure criterion that were developed for human bones (Pistoia #59). Have these been shown accurate for estimation of mouse vertebral strength?

41. Line 577: Define CCR.

42. Lines 607-...: Are there any citations to support the definitions of Synergism, Additivity and Interference? I think these are very clear and reasonable, but perhaps there is supporting literature on this?

43. The mechanical loading protocol should be described in greater detail. Additionally, did the applied 8N supraphysiological load induce any microdamage in the bone? What is the failure load of the 6th caudal vertebra?

44. In 'Pharmaceutical treatment' section, the delivery method for SOSTab is not specified and should be included.
Figures – additional comments

45. The use of 'o' in the color of corresponding group to denote statistical significance is innovative; however, it only represents $p < 0.05$ and the 'o's are very small. If there are results with more stringent significance levels (e.g., $p < 0.001$), the authors should consider denoting these separately to reflect the varying degrees of statistical strength, and consider making the symbols larger.

46. The authors should present the data with individual data points for each mouse, where applicable, to visualize the variability within groups.

47. All figure captions should clearly state what the box-and-whisker plots represent, e.g., whether the box indicates the interquartile range, the line represents the median, and the whiskers show the full range or a specific percentile.

48. Fig 5 is missing the color key for formation, resorption, and quiescence.

Minor comments

49. The manuscript primarily discusses changes in trabecular bone; however, were any changes observed in the cortical shell of the vertebral body in response to mechanical loading and/or drug treatments?

50. As Table 1 (list of statistical significances) does not provide any key additional information, this can be moved to the supplementary or consolidated with the figure. Also, a color key in the table will be helpful.

(Remarks on code availability)

Reviewer #4

(Remarks to the Author)

(Remarks on code availability)

Version 1:

Reviewer comments:

Reviewer #1

(Remarks to the Author)

Thank you for addressing my concerns; the revised version has been substantially improved and I would be fine now as it stands.

(Remarks on code availability)

Reviewer #2

(Remarks to the Author)

The authors have done a very good job at addressing my concerns with the original manuscript, which has included a new statistical approach. I think the work is much improved and congratulate them on the study.

(Remarks on code availability)

The code appears to be constructed in a way that is usable, although I have not downloaded, installed and run the application. Since I do not use R regularly, I cannot test this.

Reviewer #3

(Remarks to the Author)

The authors have diligently addressed all of my queries. The data presentation has notably improved, enhancing the clarity of the results. Overall, this impressive work provides convincing evidence Frost's mechanostat theory is a useful framework for examining treatment + loading interactions.

There are a few new/remaining comments:

1. The use of a significant interaction term to show a synergistic effect is a rigorous approach. With this approach, however, the number of outcomes that show evidence of statistical synergy are few (one outcome in Fig 2 (dStrength for Scl+ML group (Fig 2e)); four outcomes in Fig 3 (3b, 3d, 3f); two in Table S1; one in Fig S2f; six in Table S5.) Thus, I count 14 'S' for all the many comparisons and outcomes and many more 'A's. The authors have changed the Abstract to reflect that: "mechanical loading additively and synergistically enhanced predicted strength, bone volume, and mechanoregulation parameters when combined with anabolic therapies (parathyroid hormone and sclerostin antibody)..". The manuscript Title should be changed similarly to include "additive".
2. Line 111: mentioning the times of measurement at the start of this paragraph may help orient the reader to the results. "In the VEH-group, BV/TV continued to decline from w20, while all monotreatment groups showed significant increases after 4 wks treatment (p<0.001, Fig. 1b)."
3. Line 188-189: what is implication of the correlation/ no correlation of predicted strength and trabecular measurements? A summary statement of these results will be helpful.
4. Fig 2g: provide time stamps on the enlarged area in the fig. Are these w20 and 24?
5. Fig 3h: why is Δ strength vs BRR plotted as power law? Was any statistical analysis performed to determine the regression, such as Akaike information criterion?
6. Line 240: as fig 4b y labels say %SED diff, consider changing the result text to "percentage change" rather than "relative expression."
7. Fig 4c and 5a: are these representative curves (from one animal) or a curve based on average coefficients? Please clarify.
8. The methods used to determine MinR, remodeling and modeling abilities should be presented in the Methods section. While MinR is discussed in the supplementary section, it should also be included in the main text. If these parameters were calculated following a published method, please state this explicitly in the text.
9. Lines 266-267: Rather than stating that MinR represents the "basal level", might it be better described as the value where further increases in local loading stimulus do not further suppress bone resorption? "independent from mechanical signal" suggests this is the value in the non-loaded condition.
10. Figure 4c. Unclear what the dashed and solid lines indicate for the top plot (VEH). Caption legend shows that black, dashed line indicates VEH; if so what colored solid lines represent here? Is this meant to show the ML group (i.e., VEH + ML)? Same comment applies to Fig S3c.
11. Fig 4d. Please define "[-]" in the caption. (I had to look it up.)
12. Fig 4e. Where is the deltaCCR bar for BIS+MIL group?

(Remarks on code availability)

Reviewer #4

(Remarks to the Author)

(Remarks on code availability)

Response to Reviewer 1

Reviewer #1 (Remarks to the Author):

The manuscript by Schulte et al presents a novel comparison of different osteoporosis treatments in ovariectomized mice and put them in addition under mechanical compression loading. The aim of the work is to compare anabolic vs. catabolic osteoporosis treatments with or without loading (as surrogate to training intervention). Compared were bisphosphonate (BIS), parathyroid hormone (PTH), sclerostin antibody (SclAB) or vehicle (VEH) treatment in ovariectomized 20-week-old C57BL/6 mice. In clinical routine treatment of osteoporosis patients and doctors are frequently unsure if an additional physiotherapy or training would help to maintain bone mass and thus a very practical reasoning is behind the here presented study. The work is organized in a sound manner and the question targeted has very practical consequences. While I do like the approach, I do have two main concerns that could eventually be addressed by the authors:

- 1. Why did you use the selected therapies in this specific comparison? Did you formulate initially more concrete hypothesis for each individual comparison loaded vs. sham in bisphosphonate (BIS), parathyroid hormone (PTH) and sclerostin antibody (SclAB)? E.g. regarding OB or OC or synergistically? A more stringent biological explanation for the comparisons would be helpful.*

We have reformulated the hypotheses and believe that reformulation with a more stringent biological explanation of the rationale behind the selection of therapies, with each treatment targeting a different component of the bone cell network, helps strengthen the manuscript.

Changed text (Introduction, I.40-101):

Postmenopausal osteoporosis in women is characterized by increased bone resorption, leading to structural deterioration of the bone micro-architecture and increased susceptibility to fractures¹. Exercise is fundamental to bone health, but its role alongside pharmacological treatment remains an area of active inquiry². Patients with diagnosed osteoporosis often question the compatibility of exercise with pharmacological treatment, some fearing interference with drug efficacy, others assuming that their potent medications obviate the need for physical activity. This highlights the need for evidence on whether mechanical loading improves outcomes for bone strength beyond pharmacological monotherapy.

[...]

On the cellular level, anti-resorptive bisphosphonates (BIS) prevent osteoclast attachment and activity by binding to hydroxyapatite crystals in the bone matrix, and are characterized by low osteoanabolic capacity⁷. We hypothesized that mechanical loading would stimulate bone formation, but bisphosphonates might blunt this response by inhibiting (re)modeling.

Intermittent parathyroid hormone (PTH) exerts anabolic effects by binding to PTH-receptors on osteoblasts and osteocytes, leading to suppression of sclerostin expression and subsequent activation of the Wnt/ β -catenin signaling pathway. This promotes osteoblastic activity and indirectly enhances osteoclast recruitment via upregulation of RANKL expression by osteoblasts⁸. PTH and mechanical loading have

shown synergistic effects in cortical bone^{9,10}. We hypothesized that in trabecular bone, their combination would suppress otherwise increased resorption, and concurrently stimulate bone formation¹¹.

Sclerostin antibodies (SclAB) inhibit a Wnt/ β -catenin-signaling antagonist protein, promoting osteoblast activity and suppressing osteoclast activity¹². Wnt/ β -catenin-signaling is initiated when a Wnt ligand binds to a Frizzled family receptor and its co-receptor LRP5/6 on the surface of a cell, resulting in β -catenin stabilization and target gene transcription inside the cell¹¹. We hypothesized mechanical loading in combination with SclAB would increase formation and decrease resorption further. Synergy may arise because mechanical loading downregulates sclerostin expression in osteocytes¹³, while SclAB pharmacologically neutralizes it, amplifying the same pathway from two directions, especially in trabecular bone which particularly adapts its microarchitecture to mechanical demands. It should be noted that, similar to mechanical loading, sclerostin antibody effects saturate after a certain time, which limits their long-term use as a standalone therapy¹⁴.

[...]

Frost's mechanostat theory¹⁵⁻¹⁷ proposes that bone adapts to maintain mechanical strain within a defined setpoint range - much like a thermostat. The setpoints separate the four following states: the disuse state for bone resorption; the adapted state for quiescence, the overload state for formation, and the fracture state for microdamage and fractures. The mechanostat theory was introduced more than 30 years ago. In this theoretical construct, anti-catabolic bisphosphonates help inhibit increased bone resorption in underloaded areas by raising the mechanical setpoints towards higher strains. Anabolic treatments theoretically promote bone formation by lowering the mechanical setpoints required to initiate formation, provided mechanical loading is also encouraged.

[...]

Here we provide experimental insights into this interaction using *in vivo* micro-computed tomography (micro-CT) and finite-element modeling in a C57BL/6 mouse model of osteoporosis. The aim of this study was to explore whether combining pharmacological treatments with mechanical loading leads to improved pharmacological treatment outcomes, i.e. stronger bones. We tested short-term effects of BIS, PTH, and SclAB treatment with or without mechanical loading of the 6th caudal vertebra, aiming to maximize bone accrual in early treatment phases. The choice of selected therapies was based on clinical relevance and distinct modes of action in the bone (re)modeling process.

Anti-catabolic bisphosphonates help inhibit increased bone resorption in underloaded areas by raising the mechanical setpoints towards higher strains. Anabolic treatments theoretically promote bone formation by lowering the mechanical setpoints required to initiate formation, provided mechanical loading is also encouraged.

The choice of selected therapies was based on clinical relevance and distinct modes of action in the bone (re)modeling process.

Changed text (Discussion, I. 372 - 375): Here, we investigated how three current pharmacological treatments influence this regulation in the initial treatment phases in osteoporotic mice, particularly when

combined with mechanical loading. Our findings suggest that all three investigated treatments induce bone formation and inhibit bone resorption at the mechanically relevant sites,...

Changed text (Discussion, l. 381 - 384): This response may be explained by dual suppression of the activity of sclerostin, a protein secreted by osteocytes that antagonizes Wnt signaling¹¹. While mechanical loading independently suppresses sclerostin expression in osteocytes¹³, SclAB pharmacologically neutralizes any residual sclerostin¹¹.

- 2. Some treatments come with side effects; not all can be given for a similar (long) period. And not in a similar manner. This is due to substantial biological differences across the therapies selected. This aspect seems to be widely ignored; eventually focusing here on more comparable approaches and eventually only the initial effects (and not long-term consequences) might help?*

In response to your suggestion, we have added a discussion paragraph discussing the differences, anabolic windows and their potential impact on interpretation of long-term outcomes. Furthermore, we have clarified in the introduction that our analysis does not aim to directly compare long-term effects across biologically dissimilar therapies, but the focus is on short-term effectiveness. We hope this addresses your concern and improves the clarity and robustness of our interpretation.

Changed text (Abstract, l.37) ..., particularly in early treatment phases,

Changed text (Introduction, l.94 - 100): Here we provide experimental insights into this interaction using *in vivo* micro-computed tomography (micro-CT) and finite-element modeling in a C57BL/6 mouse model of osteoporosis. The aim of this study was to explore whether combining pharmacological treatments with mechanical loading leads to improved pharmacological treatment outcomes, i.e. stronger bones. We tested short-term effects of BIS, PTH, and SclAB treatment with or without mechanical loading of the 6th caudal vertebra, aiming to maximize bone accrual in early treatment phases.

Changed text (Discussion, l.435 – 439): Given the biological differences among the therapies selected, including administration routes and duration limitations, direct long-term comparisons are not the aim of this study. Instead, our focus on short-term effects aimed to investigate potential synergistic improvements in bone strength and to gain mechanistic insight into how pharmacological and mechanical stimuli modulate mechanoregulation.

- 3. If the work focuses only on osteoblasts and osteoclasts and their activities, I would assume that more in depth analyses on the tissue (histological) level could help confirm the findings from the μ CT data.*

Thank you for this valuable comment. We agree that histological analyses would provide important complementary insights to support the cell-specific interpretations derived from the micro-CT data. However, this study was specifically designed to prioritize longitudinal *in vivo* assessment of dynamic bone (re)modeling, with a focus on surface-level formation and resorption activities. Following animal sacrifice, bone samples were embedded in polymethyl methacrylate (PMMA) for synchrotron radiation-based computed tomography (SR CT). Unfortunately, this embedding protocol precluded subsequent histological processing, and some samples were lost during preparation. This is a limitation of the study and has now been

acknowledged in the revised manuscript. We have added the following statement to the Discussion section to clarify this point:

Changed text (Discussion, l.411 - 423): This study focuses on osteoblast and osteoclast surface activities using *in vivo* micro-CT, which allows longitudinal tracking of bone (re)modeling. Histological analysis could offer further insight into cellular and tissue-level mechanisms to support the cell-specific interpretations derived from the micro-CT data. However, due to the method of tissue preservation and the limited number of available samples, it was not considered feasible in the current study. It's worth noting though that previous work demonstrated a significant correlation between micro-CT-derived and histomorphometry-based MS and MAR ($R \approx 0.59 - 0.78$)^{35,36}. Furthermore, Birkhold et al.³⁷ have demonstrated the ability of longitudinal *in vivo* micro-CT to reliably quantify bone (re)modeling, i.e. separation into modeling and remodeling, in both trabecular and cortical bone. While *in vivo* micro-CT does not allow direct visualization of osteocytes, our measurements reflect the functional output of the bone cells in which osteocytes play an important role through orchestration of bone formation and resorption in response to local mechanical stimuli⁶.

4. *In my eyes very interesting findings are – specifically in the light of the authors reference to Frost – the separation that formation – rest – resorption volumes seen in the PTH + ML and SclAB + ML (Figure 4), while they do overlap in almost all other cases. So maybe Frost was correct in the end – but only in a co-stimulation with PTH or SclAB. What sets these stimuli in respect to mechano-responsiveness apart from the others?*

In response to this comment, we have revised Figure 4 to better visualize differences in remodeling site distributions across groups. To formally assess variability in site-specific responses, we performed Levene's test to compare variance in SED values between formation, quiescence, and resorption across all treatment groups. Interestingly, while the reviewer highlights PTH and SclAB as visually distinct, the statistical analysis identified significant reductions in variance specifically in the BIS + ML and ML groups, suggesting these groups may exhibit more defined mechanosensitive responses under mechanical loading. We have added this statistical result and interpretation to the revised results sections. We would like to highlight that Figures 5 and 6 now explore in more detail what differentiates the stimuli in PTH + ML and SclAB + ML with respect to mechano-responsiveness from those in other groups.

Changed text (Results, l.244 – 246): Levene's test revealed significantly smaller variances between BIS and BIS + ML and between VEH and ML ($p < 0.05$), indicative of more consistent mechanosensitive responses in these groups in the presence of mechanical loading.

Changed Figure 4 legend: Figure 1: Formation aligns with high mechanical signal, resorption with low: (a) Spatial comparison of local mechanical signal (SED at week 20) with sites of trabecular bone formation and resorption (w20-22). (b) SED at formation, resorption, and quiescent sites⁶, normalized to the group mean of quiescence to allow statistical testing with this group. The box spans the interquartile range (IQR) from the 25th to the 75th percentile. The median is shown as a line inside the box. Whiskers extend to the most extreme data points within $1.5 \times$ IQR, points beyond this range are plotted as outliers. * $p < 0.05$, *** < 0.001 using two-sided paired Wilcoxon-signed rank test with Bonferroni correction. (c) Conditional probabilities of formation and resorption as a function of SED. VEH indicated by black dotted line. Values are cropped above 40% due to low voxel count⁶. (d) Asymptotic resorption probability value (\min_R). Δ denotes change from VEH; sum of single vs. combined

treatment group means indicating antagonistic (I), additive (A) or synergistic (S) effects (e) CCR as a measure of mechanoregulated (re)modeling (f) Linear regression analysis in interval 20-22 of Δ Strength with CCR. Significant differences are depicted with a symbol in the color of the corresponding group (o: to monotreatment, ●: to combined treatment, see Supplementary Table S.1 for color encoding, $p < 0.05$).

5. *The manuscript would benefit from a clear link to biology, here specifically osteoblastic or osteoclastic activities. In one sentence, the authors mention osteocytes. But there is no reasoning given on how specifically Osteoblasts or osteoclasts are affected. Frost referred more to the osteocytes, which are here in analyses and data not at all in the focus. I am a bit surprised that the biology is not really included, even though the therapies analyzed are very distinctly different in their effects on these cell types.*

We agree that stronger biological framing helps interpret imaging-based results. As a first step, we reformulated the hypotheses in a more biological context (see comment 1). Our study relies on *in vivo* micro-CT, which captures the spatial and temporal output of cellular activity, namely bone formation and resorption surfaces, but not the cells themselves. In the mechanostat framework, osteocytes are thought to serve as the primary mechanosensors, regulating osteoblast and osteoclast behavior via molecular signaling (e.g., sclerostin). While we do not directly visualize osteocytes, our observed remodeling patterns represent the outcome of their regulatory influence. We have clarified this point in both the Introduction and Discussion sections. Moreover, we now explicitly address how the treatments used (PTH, SclAB, BIS) are expected to act on different cellular populations, as stated in comment 1. We have revised the text accordingly to better reflect the cellular context.

Changed text (Introduction & Discussion): see Reviewer 1 comment 1

6. *Finally, no other cells – such as immune or endothelial cells– are here considered. This limitation needs more discussion.*

Inserted text (Discussion, I.430 - 434): Other cell types, particularly immune cells and endothelial cells, also contribute to bone (re)modeling via inflammatory signaling and vascularization, respectively. Recent work originating from fracture healing highlights the role of immune and vascular interactions, particularly in the context of aging and mechanical loading⁴². While these cell types were not analyzed here, future multimodal approaches could complement imaging-based assessments.

*Reviewer #1 (Remarks on code availability):
no further comments*

Response to Reviewer 2

Reviewer #2 (Remarks to the Author):

This work from Schulte et al examines combined effects of experimental mechanical loading with three current therapeutic approaches for osteoporosis in an animal model. This is an important question to test, and the experimental intervention they have carried out is good. The authors report that the combination of two anabolic therapies (anti-sclerostin and intermittent PTH) with mechanical loading has a (numerically) more than additive effect on trabecular bone volume and on cortical thickness. They then carry out a secondary analysis which shows that bone formation occurs where there is high predicted mechanical signal, and bone loss where there is less mechanical signal; this is confirmatory data of a general principle of bone mechanobiology, but they take it one step further in attempting to develop a data-driven theoretical model of whether there is a change in the set-point at which bone surface activity occurs. This latter part of the paper is extremely difficult to critique as it is not very clearly described. The experiments are well planned, and the discussion section of the work is very thorough, but the data presentation is unclear and I disagree strongly with the way the authors have interpreted the combined effects of the treatments they have tested.

Major concerns:

- 1. Claims of synergy and interference: The work lays claim to a synergistic effect of mechanical loading and anabolic therapy. This is an overclaim and shows a lack of understanding of synergy. Usually, in pharmacological analysis, proof of synergy depends on showing that the effect of two treatments together is much greater than the addition of the two therapies, and is based on the idea that they work through the same pathway. In this project, they have calculated the average increase of the combined treatment, and compared it numerically to adding together the average effect of each treatment. There is no statistical analysis to determine whether these "effects" are significantly different. What is also a problem is that they use this method on parameters, such as trabecular number and trabecular thickness, which are not separable, and each contribute to trabecular bone density. This overlooks the fact that an increase in trabecular thickness is frequently associated with a reduction in trabecular number, since it can reflect a loss of thinner trabeculae, rather than a specific increase in trabecular thickness. It is not possible to test for synergy on such parameters in this context. The meaning of "interference" is not clear.*

We thank the reviewer for this important and constructive comment regarding our interpretation of synergy and interference effects. We acknowledge that our initial analytical approach was not sufficient to rigorously support claims of synergy. In response to this concern, we have substantially revised our statistical analysis. Specifically, we now assess interaction effects directly within a linear model framework in R, whereby the statistical significance and direction (positive or negative) of the interaction term between treatment and mechanical loading is used to evaluate potential synergy or antagonism. This approach allows for a formal statistical test of whether the combined effect deviates significantly from an additive model, without relying solely on descriptive comparisons of group means. To ensure robustness and appropriate interpretation of these models, we have consulted with a statistical expert with experience in

experimental design and interaction modeling. This expert has reviewed and revised the statistical approach and has been added as a co-author of the manuscript. Additionally, we have revised our language throughout the manuscript to avoid overstating claims and now describe the observed interactions more cautiously and mechanistically, without assuming pharmacological synergy in the strictest sense. We also replaced the term interference with the more common term antagonism, which we now define explicitly as a statistically significant negative interaction effect (treatment:loading), suggesting that the combined treatment response is lower than expected additive effect of the individual effects. Regarding the reviewer's concern about the interpretation of trabecular parameters, we agree that trabecular number and thickness are interrelated and should not be interpreted in isolation. We now address this in the results section and avoid attributing specific mechanistic conclusions to changes in individual parameters without considering their combined effect on trabecular bone volume fraction. We appreciate the reviewer's comments, which have helped us to improve the statistical rigor and clarity of the manuscript.

Changed co-authors' list: Friederike A. Schulte¹, **Francisco C. Marques¹**, Julia K. Griesbach¹, Claudia Weigt¹, Marcella von Salis-Soglio¹, Floor M. Lambers¹, **Clemens Kreutz²**, Michaela Kneissel², Peter J. Richards^{2,3}, Gisela A. Kuhn¹, Ralph Müller^{1*}

Changed abstract:

Bone's ability to adapt to mechanical demands is governed by mechanoregulation, the process by which cells sense and respond to mechanical stimuli to maintain skeletal integrity. In osteoporosis, increased bone resorption activity leads to structural deterioration and elevated fracture risk. While existing pharmacological therapies aim to restore bone mass to reduce fracture risk, it is unclear how they modulate mechanoregulation, especially when combined with physical interventions. Here, we investigate the joint effects of load-bearing physical and pharmacological treatment in a female mouse model of osteoporosis using longitudinal *in vivo* micro-computed tomography and computational mechanics. We demonstrate that mechanical loading additively and synergistically enhanced predicted strength, bone volume, and mechanoregulation parameters when combined with anabolic therapies (parathyroid hormone and sclerostin antibody) but not with anti-catabolic treatments (bisphosphonates). Increases in predicted strength were associated with reductions in bone resorption rates, shifts in the (re)modeling thresholds as anticipated by Frost in the mechanostat theory, and the modeling capacity of anabolic pharmacological treatments. These findings underscore the therapeutic potential of combining anabolic pharmacological therapies with load-bearing physical activity, particularly in early treatment phases, to optimize bone adaptation and fracture prevention in osteoporosis management.

Changed Figure 2 legend: Mechanical loading increases the morphological effects of anabolic and anti-catabolic treatments, contributing to increased predicted strength.

Changed text (Results, l.164 - 217):

Mechanical loading modulates structural and functional responses to pharmacological interventions

We next assessed how additional mechanical loading influenced the response to pharmacological treatment. In the BIS + ML-group, BV/TV gains were significantly less than the expected additive effects (treatment:loading: $p < 0.05$, Fig. 2a), and did not differ significantly from BIS alone (Table S.1). In contrast, anabolic treatment combined with mechanical loading (PTH + ML, SclAB + ML) resulted in additive increases, as supported by a non-significant treatment:loading interaction (Fig. 2a). Δ Tb.Th was significantly higher in all combination groups compared to monotreatments after four weeks (Fig. 2b). Δ Tb.N was preserved in BIS + ML, despite the known Tb.N loss with ML alone (Fig. 2c), suggesting BIS blunted this effect. The loss of Tb.N observed in SclAB was not rescued in SclAB + ML (Fig. 2c). Conn.D as a measure of the structural integrity and interconnectedness of the trabecular network, was preserved the most in BIS + ML, and PTH + ML (Fig. 2d). It should be noted that structural parameters such as Tb.Th and Tb.N are interdependent; apparent increases in Tb.Th may arise from the loss of complete thinner trabeculae. Synergy was found in total area (Tt.Ar), leading to additive increases in cortical area fraction (Ct.Ar/Tt.Ar) and cortical thickness (Ct.Th), found in Supplementary Fig. S.1 and Table S.4. Predicted strength increased in all treatment groups, with loading-induced increases being additive in the PTH + ML group and synergistic in the SclAB + ML group ($p < 0.05$, Fig. 2e). Strain energy density (SED) decreased in all treatment groups, being most pronounced in the combination groups, with less than the expected additive reduction in BIS + ML (treatment:loading: $p < 0.05$, Fig. 2f, Table 1). Illustrating the effect of decreasing SED, the visualization of the 6th caudal vertebra in a SclAB + ML animal shows a highly loaded trabecula at week 20, followed by structural reorganization and reduced SED at week 24 (see Fig. 2g). Linear regression analysis revealed that the changes in predicted strength did not correlate with baseline BV/TV ($R = 0.010$) and moderately with Tb.Th ($R = 0.424$, Fig. 2h).

Combination treatments affect bone formation and resorption dynamics differentially

Bone formation rate per bone volume (BFR) was lowest in VEH and highest in the anabolic combination groups ($p < 0.05$, Fig. 3a). MAR showed synergy in SclAB + ML (treatment:loading: $p < 0.05$, Fig. 3b). MS increased additively in PTH + ML and SclAB + ML, and antagonistically in BIS + ML (treatment:loading: $p < 0.01$, Fig. 3c). Bone resorption rate per bone volume (BRR) was reduced in all treatment groups ($p < 0.05$, Fig. 3d). In BIS + ML, the BRR reduction was antagonistically affected by mechanical loading (treatment:loading: $p < 0.05$). In contrast, BRR decreased synergistically in PTH+ML (treatment:loading: $p < 0.05$, Fig. 3d) and additively in SclAB + ML (Fig. 3d). MRR remained suppressed in BIS + ML and was unaffected by mechanical loading (Fig. 3e). ES decreased synergistically in PTH + ML (treatment:loading: $p < 0.05$, Fig. 3f) and SclAB + ML (treatment:loading: $p < 0.01$, Fig. 3f). This synergistic effect was, at least in SclAB + ML, still present in the second time interval (see Supplementary Fig. S.2 and Table S.5 for results from w22–24). Representative trabecular microstructures and insets of the combination groups are depicted in Fig. 3g. Changes in predicted strength correlated linearly with BFR ($R = 0.830$) and decayed according to a power-law relationship with BRR ($\rho = 0.760$), indicating that small reductions in bone resorption rates under treatment can result in large increases in predicted strength (Fig. 3h).

Figure 3 legend: Bars show group means of sum of single vs. combined treatment indicating: antagonistic (I), additive (A) or synergistic (S) effects determined via treatment:loading interaction in a linear model on the \log_2 -transformed values. Significances were calculated using ANOVA followed by Tukey's post-hoc test; significant differences are depicted with a symbol in the color of the corresponding group (○: to

monotreatment, ●: to combined treatment, $p < 0.05$). (g) Visual representations of combination groups BIS + ML, PTH + ML, and SclAB + ML in week intervals w20-22 and w22-24 (h) Linear regression of Δ Strength with BFR, and exponential regression analysis with BRR in week interval w20-22.

Changed text (Results, I.233 – 241)

Bone formation and resorption occur in mechanically distinct strain windows

To address the question whether pharmacological treatments target bone (re)modeling to the places that mechanically need it, we correlated the underlying mechanical signals in the trabecular compartment with subsequent formation and resorption events. Visually, spatial agreement was found between high SED and regions of subsequent bone formation, and low SED and regions of subsequent bone resorption (Fig. 4a). Quantitative analysis (Fig. 4b) showed that mean SED was significantly higher at formation sites ($p < 0.05$) and significantly lower at resorption sites ($p < 0.05$) compared to quiescent surfaces, when expressed relative to the average quiescent SED across all animals per group.

Changed Figure 4 legend: Δ denotes change from VEH; sum of single vs. combined treatment group means indicating antagonistic (I), additive (A) or synergistic (S) effects (e) CCR as a measure of mechanoregulated (re)modeling (f) Linear regression analysis in interval 20-22 of Δ Strength with CCR. Significant differences are depicted with a symbol in the color of the corresponding group (○: to monotreatment, ●: to combined treatment, $p < 0.05$).

Changed Figure 5 legend: Bars show group means of sum of single vs. combined treatment indicating: antagonistic (I), additive (A) or synergistic (S) effects determined via treatment:loading interaction in a linear model on the absolute values. Significances were calculated using ANOVA followed by Tukey's post-hoc test; significant differences are depicted with a symbol in the color of the corresponding group (○: to monotreatment, ●: to combined treatment, $p < 0.05$). (g) Linear regression of Δ Strength with FSL, and power-law regression analysis with RmT from week interval w20-22.

Added text (Results, I.317 - 321): Furthermore, FSL correlated strongly with predicted strength increases ($R = 0.888$), while RmT exhibited a moderate inverse correlation ($\rho = 0.609$) when fitted with a power-law function. These findings suggest that a reduction in RmT may be a prerequisite for strength gains; however, actual increases in strength require anabolic potential (Fig. 5e,f).

Changed text (Discussion, I.369 - 381):

The main goal of osteoporosis therapy is to restore bone mass to reduce fracture risk. To achieve this efficiently, existing bone must be preserved and new bone formed at mechanically relevant sites. In healthy bone, bone (re)modeling is proposed to be regulated via the mechanostat^{6,19-21}. Here, we investigated how three current pharmacological treatments influence this regulation in the initial treatment phases in osteoporotic mice, particularly when combined with mechanical loading. Our findings suggest that all three investigated treatments induce bone formation and inhibit bone resorption at the mechanically relevant sites, but it was only in anabolic therapies that mechanical loading additively (PTH) or synergistically (SclAB) enhanced predicted bone strength.

When combined with sclerostin antibody treatment, mechanical loading led to a synergistic increase in predicted bone strength and an additive increase in bone volume fraction. Mineral apposition rates and reductions in eroded surfaces were synergistic, consistent with previous findings^{12,22,23}.

Changed text (Discussion, I.392 - 394): We found additive increases in predicted strength and increased sensitivity of mechanoregulation mechanisms for PTH + ML, which is in line with previous results where PTH increased the mechanoregulation of osteoblasts only when combined with fluid flow²⁹.

Changed text (Discussion, I.400 - 406): Bisphosphonates were the only treatment that preserved Tb.N but showed no additive effects with mechanical loading, consistent with clinical^{32 33} studies reporting limited benefit of combining bisphosphonates and exercise^{32 33}. Since mineralizing surface was significantly increased in the bisphosphonate group compared to the vehicle group, our results suggest that bisphosphonates not only affect the activity of osteoclasts but also that of osteoblasts, possibly through disruption of the coupling process via elevated osteoclast apoptosis³⁴, potentially impairing the spatial guidance of osteoblasts by the mechanical signal.

Changed text (Discussion, I.439 - 457): The findings support mechanostat theory: treatments lower mechanical setpoints, enabling bone formation at lower strains. However, only anabolic treatments effectively reinforce highly loaded trabeculae and reduce peak strains, underscoring the importance of modeling capacity for strength gains. And while our results highlight the promise of mechanical loading as an adjunct to anabolic therapies, clinical studies will ultimately be required to determine optimal exercise modalities, just as pharmacological dosing requires careful optimization.

In conclusion, the current study revealed that drug treatments target trabecular bone (re)modeling at mechano-functionally relevant places, and osteoanabolic treatments in combination with mechanical loading additively and synergistically increased the bone response. The additive (PTH) and synergistic (SclAB) increases in predicted strength can be explained by reductions in resorption rates, shifts in the (re)modeling thresholds as described by Frost in the mechanostat theory more than 30 years ago, and by the modeling ability of anabolic treatments. Being able to link organ-level changes induced by physical or pharmacological therapies to the mechanosensitivity of the cells provides information on how well skeletal structures can adapt to mechanical demands under pathological or treatment conditions, a task for which the body has optimized itself over millions of years of evolution. Load-bearing exercise may represent a promising adjunct option for physically active osteoporosis patients on osteoanabolic therapy.

Changed text (Methods, I. 583 - 630)

Statistical methods

Statistical analysis was performed using R (R, Auckland, New Zealand⁵³). Data are presented as mean ± standard error unless otherwise specified. P-values lower than 0.05 are considered significant, statistical significance was defined as *p<0.05, **p<0.01, ***p<0.001. Normality was assessed using the Shapiro–Wilk test (in R, with $\alpha = 0.05$).

Static morphometric and mechanical parameters of trabecular bone in the 6th caudal vertebra were measured at weeks 15, 20, 22, and 24, with Δ representing the percentage change between weeks 20 and 24. ANOVA followed by Tukey's post-hoc test was used to compare Δw_{20-24} . Synergy (S) or antagonism (I) between

treatment and mechanical loading was evaluated using the interaction term ("treatment:mechanical loading") in a linear model.

A synergistic effect was defined as a combined treatment response that exceeded the sum of the individual effects of treatment and mechanical loading. This implies that the two interventions may act through the same or facilitating biological pathways. Statistically, such synergy is indicated by a significant and positive interaction term between the main effect (treatment) and mechanical loading. An antagonistic effect was defined as a combined treatment response that was less than the expected additive effect of treatment and mechanical loading. Statistically, such antagonism was reflected by a significant and negative interaction term between the main effect (treatment) and mechanical loading. The animal provider from the third (SclAB) and fourth (PTH) experiment was different, therefore we adjusted the model for this.

Dynamic morphometric parameters were analyzed during weeks 20-22 and 22-24. ANOVA followed by Tukey's post-hoc test was applied to absolute values in each time interval. Synergy or antagonism was assessed using the interaction term from a linear model of log₂-transformed values in each time interval.

Strain Energy Density (SED) in formation (F), quiescence (Q), and resorption (R) surfaces was reported for each group in each time interval. Differences between F, Q, and R were compared using two-sided Wilcoxon signed-rank tests with Bonferroni correction after testing for normality.

For conditional probabilities, exponential curves were fitted to the formation and resorption slopes. For formation, a saturating exponential model $f(x) = y_0 + a \cdot (1 - \exp(-bx))$ was applied to the first 40% of normalized SED data, with parameters y_0 , a , and b representing baseline, maximal formation, and sensitivity to mechanical stimulus, respectively. For resorption, a decaying exponential model $f(x) = y_0 + a \cdot \exp(-bx)$ was used to assess the rate of resorption. Synergy or antagonism was assessed using the interaction term from a linear model applied to log₂-transformed coefficients, with significance determined by ANOVA followed by Tukey's post-hoc test.

Formation and resorption saturation levels (FSL, RSL) and remodeling thresholds (RmTs) were derived from hyperbola remodeling velocity curves, fitted with the function $f(x) = y_0 + \frac{a}{(b-x)}$, where x represents the mechanical signal, y_0 the FSL, and a the (re)modeling velocity modulus. The remodeling threshold (RmT) is defined as: $RmT = b + \frac{a}{y_0}$. ANOVA followed by Tukey's post-hoc test was used for statistical comparisons, with synergy or antagonism assessed using the interaction term from a linear model applied to absolute values in each time interval, because FSL and RSL can obtain negative values.

Linear correlations between the %-change in predicted strength and parameter outcomes were assessed using Pearson's correlation coefficient R . For nonlinear relationships, we employed a power-law model of the form $y = a \cdot x^b$, which was linearized via logarithmic transformation to $\ln(y) = \ln(a) + b \cdot \ln(x)$. Parameters a and b were estimated using linear regression on the transformed data, and the Spearman's correlation coefficient ρ between the log-transformed observed and predicted values was used to assess the goodness of fit.

Changed text (Author contributions, I.641 - 645):

FAS, FCM, JG, CW, MS, FML, GAK performed image processing and analysis. MK provided pharmacological agents. RM, GAK, PJR, CK supervised the project. FAS wrote the original draft of the

manuscript. All authors contributed to data interpretation, were involved in discussions throughout the study, and participated in manuscript revision and preparation.

Changed text (Competing interests, I. 649)

MK, PJR, and CK are employees of Novartis.

2. Data presentation: The presentation of much of the data as percentage change from baseline obscures the observations being made here; the data is not at all transparent, though I appreciate that an in vivo study allows you to see changes over time. Inclusion of the raw data in Supplementary Figures is needed for data transparency.

The raw data are now included in Supplementary Tables. Also, where appropriate, we show changes in the figures in the original units (dynamic bone morphometry and mechanoregulation analysis).

Changed Figures: Figure 3- 6

Added: Supplementary Tables 1-9

Added text (Results):

I.117: A full summary of these microstructural parameters is provided in Supplementary Table S.2.

I.161, 229, 259, 362: see Supplementary Table S.1 for color encoding, $p < 0.05$.

I.169: alone (Supplementary Table S.2). In contrast,

I.134: see also Supplementary Table S.5 for results from the second interval, w22–24),

I.180: (Ct.Th), found in Supplementary Fig. S.1 and Table S.4. Predicted

I.212: Supplementary Fig. S.2 and Table S.5 for results from w22–24). Representative

I.263: (Fig. 4c, Supplementary Fig. S.3 and Table S.6 for results in w22-24).

I.269: Supplementary Tables S.7 and S.8.

I.299: FSL was increased in all treatments compared to VEH ($p < 0.001$, Fig. 5b, Supplementary table S.9)

I.308: compared to PTH alone ($p < 0.05$, Supplementary Table S.9).

I.313: These results were even more pronounced in the second time interval (Supplementary Fig. 4d, Table S.9).

I.485: would have increased data complexity (see Supplementary Table S.3).

3. It seems that three different vehicle groups have been combined into a single vehicle group of samples. This is not good practice, and will have a major effect on the statistics for this study. The vehicle groups should be shown separately, at least to confirm that there was no difference between the three different vehicle groups.

The VEH groups were drawn from the same underlying population and conducted under identical experimental conditions at different time points; they were therefore combined to improve clarity and interpretability of the results, as separate presentation would not yield additional insights but substantially increase data complexity. Nevertheless, we understand the reviewer's concern and now provide a table in the Supplementary Material which lists the absolute and %-change values in bone volume fraction in the three vehicle groups (Table S.3), showing that significant differences in absolute values exist in weeks 15 and 24. No significant differences were found in %-change values. In consultation with the statistics expert we therefore test synergy on the change in bone response. We include the information about the animal provider and adjust for this in the statistical evaluation ($\text{lm}(\text{Diff} \sim \text{group} \times \text{treatment} + \text{provider})$).

Changed text (Methods, I.600 - 601): The animal provider from the third (SclAB) and fourth (PTH) experiment was different, therefore we adjusted the model for this.

Added table: Supplementary Table S.2.

Changed text (Methods, I.479 - 486): The bone responses of the VEH and the ML groups of both mouse lines (JRccHsd and 6JRj) showed no differences in reactions ($\Delta\text{BV}/\text{TV}_{\text{w}20-24}$; VEH: $p = 0.937$; ML: $p = 0.206$; two-way ANOVA followed by Tukey's post-hoc test), thus ML groups from all experiments and VEH groups from all experiments were merged, explaining the larger sample size in these groups. The merging was conducted because separate vehicle group presentations would not have yielded additional insights but substantially would have increased data complexity (see Supplementary Table S.3).

4. *It is surprising that, in the single agent treatment groups, that there was a reduction in BRR/BV and ES with PTH treatment, and a reduction in MAR with anti-sclerostin. This needs to be discussed.*

This response happens in the second time interval more pronounced than in the first time interval. As more questions arose regarding the time intervals presented (see Reviewer 3, comment 4), we decided to show the first instead of the second time interval of the bone response. PTH has a higher response in the first time interval. ES is not significantly different in this time interval from VEH. Still, bone resorption rate BRR is lower than VEH but not as low as BIS and SclAB. We address this now in the text.

Changed text (Results, I.130 - 142): Bone resorption rate per bone volume (BRR) was significantly reduced in all treatment groups versus VEH (Fig. 1j), where BRR in PTH was the least reduced (significantly higher than VEH, ML, and BIS), consistent with the known anabolic effects of intermittent PTH treatment. Surprisingly, BRR was found to be significantly lower than VEH, while not as low as in ML and BIS. This decrease, even more pronounced in the second time interval (see Supplementary material Table S.5), may result from micro-CT missing parts of the reversal phase between resorption and formation, thus potentially underrepresenting remodeling at the same sites. As expected, mineral resorption rate (MRR) was significantly lower in the BIS-group compared to VEH and unchanged in the anabolic groups (Fig. 1k), suggesting bisphosphonates preserve Tb.N (Fig. 1d) by preventing full trabecular resorption. Eroded surface per bone surface (ES), reflecting the proportion of bone surface previously affected by resorption, was significantly lower in ML and SclAB compared to VEH, BIS, and PTH ($p < 0.05$, Fig. 1l).

5. *The paper is not written for a general audience, apart from the introduction. The results section is highly technical and full of jargon, making it extremely challenging to follow. What is the "natural law" that you refer to in line 250, for example? I found it almost impossible to understand the final two figures, and their description on the results section, despite being very interested in the outcomes. Part of this is the continuing mis-use of the term "synergy", but also these sorts of highly theoretical methods need to have some clear explanation for non-theorists, particularly to justify the mathematical approach being used.*

We have revised the results sections to improve clarity with a short concluding summary in the end of each section. Figures 5 and 6 have been substantially updated, and we removed the term "natural law" from the text. We appreciate the reviewer's comment on the use of the term 'synergy' and agree that it requires a clearer definition. To address this, we have added a paragraph in the Statistical Methods section that formally defines synergy, distinguishes it from antagonism, and explains our choice of mathematical interaction modeling for readers less familiar with theoretical approaches. Additionally, we have introduced new subheadings in the Results section to help summarize key findings for non-specialist readers.

Changed headlines (Results):

"Micro-architectural and (re)modeling differences reveal treatment-specific effects on bone recovery after ovariectomy"

"Mechanical loading modulates structural and functional responses to pharmacological interventions"

"Combination treatments affect bone formation and resorption dynamics differentially"

"Frost's mechanostat setpoints are shifted towards lower values for mono- and combined treatments"

"The amount of modeling correlates highly with increases in predicted strength for all treatments"

Changed figures: Figure 5, 6

Changed figure legends:

Figure 5: (Re)modeling velocity curves as an approximation of the mechanostat theory. (a) (re)modeling velocity (RmV) curves for VEH / ML, BIS / BIS + ML, PTH / PTH + ML, SclAB / SclAB + ML; (b) formation saturation level (FSL) (c) resorption saturation level (RSL), (d) (re)modeling threshold (RmT) **(b-d):** The box spans the interquartile range (IQR) from the 25th to the 75th percentile. The median is shown as a line inside the box. Whiskers extend to the most extreme data points within $1.5 \times$ IQR, points beyond this range are plotted as outliers. Δ denotes difference from VEH as no baseline value from week 20 was available. Bars show **group means** of sum of single vs. combined treatment indicating: antagonistic (I), additive (A) or synergistic (S) effects **determined via treatment:loading interaction in a linear model on the absolute values.** Significances were calculated using ANOVA **followed by Tukey's** post-hoc test; significant differences are depicted with a symbol in the color of the corresponding group (**o**: to monotreatment, **●**: to combined treatment, $p < 0.05$). **(g)** Linear regression **of Δ Strength with FSL,** and **power-law** regression analysis with **RmT from week interval w20-22.**

Figure 6: Modeling capacity correlates highly with the increase in predicted strength. (a) Heterogeneity of the strain distribution decreases to different extents in different treatments. Qualitative comparison of changing SED-distribution for VEH, BIS + ML, PTH + ML and SclAB + ML over the course of 4 weeks of treatment. **(b)** Remodeling as calculated from formation following resorption at the same location over 3 measurement time points (w20, w22, w24). **(c)** Modeling as calculated from formation following quiescence at the same location over 3 measurement time points (w20, w22, w24) **(b-c):** The box spans the interquartile range (IQR) from the 25th to the 75th percentile. The median is shown as a line inside the box. Whiskers extend to the most extreme data points within $1.5 \times$ IQR, points beyond this range are plotted as outliers. Δ denotes change from VEH; sum of single vs. combined treatment group means indicating graphically antagonistic (I), additive (A) or synergistic (S) effects. Significant differences are depicted with a symbol in the color of the corresponding group (\circ : to monotreatment, \bullet : to combined treatment, $p < 0.05$). **(d)** Remodeling amount in % correlated to increases in predicted strength fitted using a power-law function. **(e)** Modeling amount in % correlated to increases in predicted strength fitted using linear regression. **(f)** Surface event proportion (modeling or remodeling) in relation to the mechanical signal. **(g)** Spider plots summarizing the mean of outcome parameters of the analyzed treatments. Values were standardized to facilitate visual comparisons between groups.

Added text (Results, l.291 - 350):

Frost's mechanostat setpoints are shifted towards lower values for mono- and combined treatments

To assess potential shifts in the setpoints due to different drug or combination treatments, (re)modeling velocity (RmV) curves and mechanostat parameters¹⁹ were calculated for each treatment group as an approximation to the mechanostat theory. These curves show the net RmV for a given surface effective strain, where a negative value means net resorption and a positive value means net formation (Fig. 5a). From these curves, formation saturation levels (FSL), resorption saturation levels (RSL), and (re)modeling thresholds (RmTs) were derived.

FSL was increased in all treatments compared to VEH ($p < 0.001$, Fig. 5b, Supplementary table S.9), and was significantly higher in anabolic combination groups compared to the respective monotreatments ($p < 0.001$). This suggests an increased mechanosensitivity of the cells involved in bone formation. RSL was increased in the BIS and all combination groups compared to VEH ($p < 0.05$, Fig. 5c). The increase in RSL under bisphosphonate treatment may be explained by the suppression of bone turnover: at comparable strain levels, resorption would typically occur, resulting in a lower RSL; however, due to inhibited osteoclast activity, this resorptive response is blunted, making the system appear saturated at a higher RSL value. Regarding the role of additional mechanical loading, only the PTH + ML group showed a significant increase compared to PTH alone ($p < 0.05$, Supplementary table S.9). This suggests that, under strain conditions that would typically lead to bone resorption, the balance has shifted towards bone formation. RmT values, representing the setpoints in the mechanostat theory, were decreased in all pharmacological treatment groups compared to VEH ($p < 0.05$), with no significant improvements observed in the combination groups receiving mechanical loading (Fig. 5d). These results were even more pronounced in the second time interval (Supplementary Fig. 4d, Table S.9). The reduction in RmT suggests an upregulation of osteocytic mechanosensitivity, an expected outcome in the anabolic treatment groups, but

an unexpected finding in the bisphosphonate-treated groups. This may reflect a compensatory osteocytic response to the suppressed bone resorption induced by bisphosphonate treatment. Furthermore, FSL correlated strongly with predicted strength increases ($R = 0.888$), while RmT exhibited a moderate inverse correlation ($\rho = 0.609$) when fitted with a power-law function. These findings suggest that a reduction in RmT may be a prerequisite for strength gains; however, actual increases in strength require anabolic potential (Fig. 5e,f).

The amount of modeling correlates highly with increases in strength for all treatments

Next, we distinguished between the remodeling and the modeling ability of the different treatments. Visually, in several areas in VEH, high SED values decreased over time, but in other areas they increased and the resorption of a single trabecula led to a more heterogeneous SED distribution (Fig. 6a). While SED on average decreased over time for BIS + ML, this did not particularly affect areas with high SED, as there remained several highly strained areas in week 24. In contrast, both PTH + ML and SclAB + ML showed large decreases in SED, especially in areas with high SED values in week 20 (Fig. 6a).

The remodeling ability - as represented by the amount of resorption followed by formation - was in BIS, BIS + ML, PTH and SclAB comparable to VEH, and significantly lower in ML and the anabolic combination groups compared to VEH ($p < 0.05$, Fig. 6b). In contrast, the modeling ability was significantly higher compared to VEH in all but the BIS group, with the highest amount in the anabolic combination treatments ($p < 0.001$, Fig. 6c). Both remodeling and modeling capacity showed antagonistic effects in BIS + ML and additive effects in anabolic combination treatments (Fig. 6b,c). Remodeling showed a moderate inverse correlation with increases in predicted strength when fitted with a power-law function ($\rho = 0.476$, Fig. 6d), while modeling correlated strongly in a linear fit ($R = 0.928$, Fig. 6e). Fig. 6f illustrates the distribution of surface events relative to increasing effective strain. Remodeling events appeared strain-independent, and modeling increased with higher mechanical stimuli, particularly in the anabolic combination groups (Fig. 6f). There is also a non-linear increase in modeling with mechanical loading in comparison to the mono-treatment, with a stronger effect for low effective strain regions, especially for SclAB. Fig. 6g summarizes the trabecular bone response across treatments. Bisphosphonates, alone or with mechanical loading, help preserve trabecular interconnectedness. Anabolic treatments combined with mechanical loading effectively enhance predicted strength by promoting formation and suppressing resorption. These additive effects across parameters likely underlie the observed synergistic gains in predicted strength, at least in SclAB.

Deleted text:

p.13, l.252 (old; Results): "of this apparent natural law".

p.13, l.254 (old; Results): "It could, however, be that the natural law is that strong that it masks subtle differences between treatments."

Changed figure: 6

Changed Figure 6 legend: **Figure 6: Modeling capacity correlates highly with the increase in predicted strength. (a)** Heterogeneity of the strain distribution decreases to different extents in different treatments. Qualitative comparison of changing SED-distribution for VEH, BIS + ML, PTH + ML and

ScIAB + ML over the course of 4 weeks of treatment. **(b)** Remodeling as calculated from formation following resorption at the same location over 3 measurement time points (w20, w22, w24). **(c)** Modeling as calculated from formation following quiescence at the same location over 3 measurement time points (w20, w22, w24) **(b-c)**: The box spans the interquartile range (IQR) from the 25th to the 75th percentile. The median is shown as a line inside the box. Whiskers extend to the most extreme data points within $1.5 \times$ IQR, points beyond this range are plotted as outliers. Δ denotes change from VEH; sum of single vs. combined treatment group means indicating graphically antagonistic (I), additive (A) or synergistic (S) effects. Significant differences are depicted with a symbol in the color of the corresponding group (\circ : to monotreatment, \bullet : to combined treatment, $p < 0.05$). **(d)** Remodeling amount in % correlated to increases in predicted strength fitted using a power-law function. **(e)** Modeling amount in % correlated to increases in predicted strength fitted using linear regression. **(f)** Surface event proportion (modeling or remodeling) in relation to the mechanical signal. **(g)** Spider plots summarizing the mean of outcome parameters of the analyzed treatments. Values were standardized to facilitate visual comparisons between groups.

Changed text (Methods, 1.557 - 581):

Local mechanoregulation analysis

The local mechanical environment was derived from micro-FE simulations of the baseline scan. Here, SED is used as a mathematical term to describe the (re)modeling stimulus phenomenologically. To quantify the relationship between the local SED and cellular activity on the bone surface, the bone formation or resorption sites determined by rigid registration were projected onto the surface of the baseline scan, resulting in three masks representing three different clusters of formed (F), quiescent (Q) or resorbed (R) bone, as described previously⁶. In short, the mean SED value was calculated in each of these three masks and for each mouse, and normalized to the mean SED of the quiescent surface as absolute SED values differ per mouse and over time due to differences in BV/TV. To establish a quantitative description of the mechanoregulatory system, the relation between increasing mechanical stimuli and consequent (re)modeling events was assessed by calculating the relative percentage of voxels being formed, quiescent and resorbed for each value of SED binned at 1% step size of the maximum SED, and assuming that each (re)modeling event has the same occurrence probability (i.e., formation, resorption and quiescent regions were virtually rescaled to have the same amount of voxels) to remove dependencies on imbalances in the (re)modeling process due to differences between animals or treatments. The conditional probabilities were fitted per animal by exponential functions. CCR values were calculated from the mean conditional probabilities from all animals, returning a scalar value between 0.33 and 1 assessing overall mechanoregulation, with 0.33 meaning (re)modeling is independent of the mechanical signal and 1 meaning it is entirely dependent on the mechanical signal. (Re)modeling velocities (RmVs) and their mechanostat parameters were calculated according to Marques et al. for each animal¹⁹. Minimum and maximum RmV, represented as formation saturation level (FSL) and resorption saturation level (RSL), as well as (re)modeling thresholds (RmTs) were extracted from the resulting curves.

6. Minor points

The authors make many comments in the introduction and discussion which are not supported by references (e.g line 58, 78, 82...)

Inserted reference (old: I.58; new: I.79): ...which limits their long-term use as a standalone therapy¹⁵." (Rauner et al. 2021)."

Deleted text and reference (old: I.78): ...allows microdamage to accumulate²⁰, ultimately shifting... (Allen et al. 2017)

Deleted text and reference (old: I.82): "...could enhance the effectiveness of therapies by promoting bone adaptation in a more dynamic, functional way¹⁹." (Frost, 2003)

Inserted references (Discussion, I.379 - 388):

Mineral apposition rates and reductions in eroded surfaces were synergistic, consistent with previous findings^{12,22,23}. This response may be explained by dual suppression of the activity of sclerostin, a protein secreted by osteocytes that antagonizes Wnt signaling¹¹. While mechanical loading independently suppresses sclerostin expression in osteocytes¹³, SclAB pharmacologically neutralizes any residual sclerostin¹¹. Together, these effects enhance Wnt/ β -catenin activity, and likely contribute to the pronounced anabolic response²⁴. Additional pathways may also be involved, as suggested by studies in SOST-KO mice, which still showed loading-induced bone formation²⁵. One candidate is Dickkopf-1 (Dkk1), another Wnt inhibitor²⁶⁻²⁸. Robling et al.¹³ reported that mechanical loading reduced both *Sost* mRNA (-73%) and *Dkk1* (-49%) transcripts.

7. *The introduction to this paper reads more like an opinion piece rather than a critical review of the literature. This needs attention.*

We have rewritten the introduction with focus on critical review of the literature. Specifically, references were inserted more rigorously. The introduction is now focused more on more stringent biological background which led to reformulation of the hypotheses as requested by reviewer 1. The introduction was shortened as requested by reviewer 3. We hope these changes meet the reviewer's expectations.

Inserted references: see Reviewer 2, comment 6.

Text changes (Introduction): see Reviewer 1, comment 1.

8. *Some references are not appropriate - for example, the peculiar term (re)modeling (which is common, but confusing) was not used in reference 9. Also the references used in line 93 do not really support the statement being made here.*

The reference was replaced. The authors would like to point out that the original reference (Huiskes et al. 2000) proposed that modeling and remodeling could be of the same origin. When publishing previous papers with this imaging method (Schulte et al., PLoSOne, 2013), we had reviewer requests to use the term (re)modeling when describing bone formation and resorption. Please also see comment 9 for further explanation.

Deleted reference (old; p.5, I.92): Huiskes et al. 2000

Inserted reference (old; p.5, I.92): Schulte et al. 2013

9. *The use of the term (re)modeling is not really helpful in this manuscript - it is not clear whether you are talking about modeling or remodeling or simply bone surface activity. If you cannot distinguish between the two with this method, I think it is better to say "bone surface activity".*

We appreciate the reviewer's concern regarding the clarity of the term (re)modeling. We have previously received requests to change the terms we used before to this terminology and would very much prefer to be consistent with publications in this field (Schulte et al. PloSOne, 2013, Birkhold et al. Bone, 2015, Marques et al., 2023, Shyu et al., Front. Med. Eng, 2025). In the meantime, new quantification parameters have followed the nomenclature of (re)modeling, e.g. Marques et al. 2023, "(re)modeling velocity". To address the reviewer's concern, we have now clarified in the manuscript what is specifically meant by (re)modeling. Additionally, we explicitly present the differences between modeling and remodeling within the treatment groups, using methods established in prior studies that employed micro-computed tomography to distinguish these processes (e.g. Birkhold *et al.*, Bone, 2015).

Changed text (Introduction, I.54 - 58): To study these dynamic processes *in vivo*, high-resolution imaging techniques such as micro-computed tomography (micro-CT) are commonly used. While micro-CT does not resolve individual cells or molecular signals, it captures the net microstructural changes resulting from formation and resorption, which are collectively referred to as bone (re)modeling⁶.

Changed text (Results, I.323 - 350):

The amount of modeling correlates highly with increases in strength for all treatments

Next, we distinguished between the remodeling and the modeling ability of the different treatments. Visually, in several areas in VEH, high SED values decreased over time, but in other areas they increased and the resorption of a single trabecula led to a more heterogeneous SED distribution (Fig. 6a). While SED on average decreased over time for BIS + ML, this did not particularly affect areas with high SED, as there remained several highly strained areas in week 24. In contrast, both PTH + ML and SclAB + ML showed large decreases in SED, especially in areas with high SED values in week 20 (Fig. 6a).

The remodeling ability - as represented by the amount of resorption followed by formation - was in BIS, BIS + ML, PTH and SclAB comparable to VEH, and significantly lower in ML and the anabolic combination groups compared to VEH ($p < 0.05$, Fig. 6b). In contrast, the modeling ability was significantly higher compared to VEH in all but the BIS group, with the highest amount in the anabolic combination treatments ($p < 0.001$, Fig. 6c). Both remodeling and modeling capacity showed antagonistic effects in BIS + ML and additive effects in anabolic combination treatments (Fig. 6b,c). Remodeling showed a moderate inverse correlation with increases in predicted strength when fitted with a power-law function ($\rho = 0.476$, Fig. 6d), while modeling correlated strongly in a linear fit ($R = 0.928$, Fig. 6e). Fig. 6f illustrates the distribution of surface events relative to increasing effective strain. Remodeling events appeared strain-independent, and modeling increased with higher mechanical stimuli, particularly in the anabolic combination groups (Fig. 6f). There is also a non-linear increase in modeling with mechanical loading in comparison to the mono-treatment, with a stronger effect for low effective strain regions, especially for SclAB. Fig. 6g summarizes the trabecular bone response across treatments. **Bisphosphonates, alone or with mechanical loading, help preserve trabecular interconnectedness. Anabolic treatments combined with mechanical loading effectively enhance predicted strength by promoting formation and suppressing resorption. These additive effects across parameters likely underlie the observed synergistic gains in predicted strength, at least in SclAB.**

10. Use of the term "pharmaceutical" is inaccurate - it refers to pharmacological agents used in humans. The correct term here would be "pharmacological".

The word "pharmaceutical" has been replaced by "pharmacological" throughout the manuscript.

11. The inclusion of data in both the text and the figures is very confusing. It is better to simply report the data, and allow the reader to look at the figures.

We have changed the results text. As few as possible data are shown in the text, and the data are now listed in the Supplementary Material. See also Reviewer 3, comment 1.

Changed text (Results, I.124 - 133): The micro-architectural changes resulted from the time course of formation and resorption (Fig. 1f). In week interval w20-22, bone formation rate per bone volume (BFR) was significantly increased in ML and PTH compared to VEH, with no significant increases in BIS or SclAB (Fig. 1g). Mineral apposition rate (MAR) did not differ between groups (Fig. 1h), while mineralizing surface per bone surface (MS) was elevated in all treatment groups compared to VEH ($p < 0.05$, Fig. 1i). Bone resorption rate per bone volume (BRR) was significantly reduced in all treatment groups versus VEH (Fig. 1j), where BRR in PTH was the least reduced (significantly higher than VEH, ML, and BIS), consistent with the known anabolic effects of intermittent PTH treatment.

12. How were the doses of bisphosphonate, PTH and Anti-sclerostin selected?

Changed text (Methods, I.514 - 519): BIS and SclAB doses were selected based on established protocols in the literature^{24,47}. The PTH dose was determined in a dose-response study comparing 10 $\mu\text{g}/\text{kg}/\text{d}$ and 40 $\mu\text{g}/\text{kg}/\text{d}$, following an initial 80 $\mu\text{g}/\text{kg}/\text{d}$ dose proposed in the literature⁴⁸ that led to excessive increases in bone volume fraction (BV/TV) by the end of the experiment. The 40 $\mu\text{g}/\text{kg}/\text{d}$ dose was chosen to obtain BV/TV increases comparable to those observed with BIS and SclAb treatments, enabling a more balanced comparison across treatment groups (see Fig. 1b).

Reviewer #2 (Remarks on code availability):

I did not see any supporting code.

Changed text:

Code availability

All data and code are available in a Zenodo repository: 10.5281/zenodo.17256047. Please find below the link for reviewer access:

<https://zenodo.org/records/17256047?token=eyJhbGciOiJIUzUxMiJ9.eyJpZCI6IjAzODM0ZWZlYzZlYtNDY1ZS1iMTE4LWY3MzI1NGEzM2ZhMyIsImRhGEiOnt9LCljYyYw5kb20iOiIyMWRkYzAzMjZlZWNIOTBmMjVjNmJjYTFiYjAzNmY3YSJ9.snQnJFbpWzCD3wEEpFu7XIUVvnOw0Bl6v-eQUJZi97M6T9pPTnnPKKG1KzKxN4UWNbONbqFFM4YzKCyRZ42GOQ>

Response to Reviewer 3

Reviewer #3 (Remarks to the Author):

Overview of Key Results and Significance

The aim of the manuscript was to evaluate the interaction between mechanical loading and osteoporosis drug treatments in the vertebral trabecular bone of osteoporotic female mice, and test whether mechanical loading acts additively or synergistically to improve drug treatment outcomes. The authors have developed, over several years, innovative biomechanical, imaging and statistical modeling approaches that allow them to address this topic in a detailed and rigorous manner. They have selected three relevant osteoporosis drugs (two anabolic, one anti-catabolic), alone and in combination with mechanical loading, which enhances the generalizability of the findings. The main finding is that combining mechanical loading with anabolic osteoporosis drug therapy synergistically enhances bone volume and bone (re)modeling, whereas no such effects were observed with anti-catabolic drug. In addition to addressing this important topic (ie, effects of drugs + loading), the authors have framed the work in the context of the influential "mechanostat model" proposed by Harold Frost. Frost based the model largely on observations and inference (and little data). Herein, the authors have used their valuable data set to provide a rigorous test of the model, and show that its basic tenets align well with data and are indeed a useful framework for explaining (and predicting) how bone responds to loading, and how osteoporosis drugs shift this response. Further, the data provided by this manuscript is an important addition to the literature as it addresses the clinically relevant question about the role of physical therapy in osteoporotic skeletons undergoing pharmaceutical treatments.

There are issues with the clarity and presentation of results, figures, etc., which make parts of the paper challenging to follow. Most of the comments below highlight these issues.

Major Comments:

- 1. The results section can be difficult to follow, especially with the frequent inclusion of mean \pm SD values for all the groups (which is redundant with same data shown in Figures). Consider stating the % difference between groups, rather than listing the mean/SD values.*

As we are now providing the raw data as well as %-differences in the Supplementary Material, we removed them from the text in the results section.

Changed text: See Reviewer 2, comment 11.

- 2. The results include many statements of difference where it is unclear if the differences are supported by statistical testing. Please use the word 'significant(ly)' to clarify (e.g., lines 120-122).*

We have inserted the word "significant" or "($p < 0.05$)" to clarify significance throughout the results section. Significance levels are now found in Supplementary Material.

3. *What is the rationale for limiting the SED/SEDmax range to 0.0-0.4 (eg, Fig 5)?*

Added text (Figure 4 legend): *Values are cropped at 40% of the full interval due to the small number of voxels above this threshold as was described previously¹⁹.*

4. *The in vivo micro-CT was done at 20, 22 and 24 wks age. It is not always clear which time interval is used for calculation of dynamic indices – please add this information as a label on the relevant figures. (It is stated in the captions but is hard to find.) If different intervals are used in different sections of the Results, state the rationale and if the results are sensitive to the interval used. This is an important concern that must be addressed. For example, in Fig 1 caption it indicates w22-w24 was used to determine remodeling rates in panels g-l. Figure 3 does not mention the interval for panels a-f, and in Figure 4, it indicates w20-w22 was the interval used to determine formed/quiescent/resorbed bone. Please justify the use of interval used, and state clearly in the figure captions.*

Thank you for raising this issue. We provide now the raw data in the Supplementary material in all time intervals. In the figure captions and in the figure panels, we state explicitly (as a subscript of each readout) which time interval is shown. For consistency, we now also show the first time interval in figure 1.

Changed figures: Figure 1-6, S.1-S.4

Changed figure 1 legend: week interval 20-22 (before: 22-24)

5. *Addition of supplemental data would enhance the rigor of the paper.*

The authors acknowledge this comment. Supplementary material containing raw data is now provided. The addition of Supplementary material also helped in addressing several of the other reviewer comments.

Added document: Supplementary material.docx

Changed text: Please see Reviewer 2, comment 2.

6. *Specific Comments:*

Introduction

The Introduction could benefit from additional details about the PTH analog and bisphosphonate, similar to the level of detail provided for the sclerostin-targeting drug.

We provide now additional details about PTH and bisphosphonate to be consistent amongst the three investigated drugs.

Changed text: see Reviewer 1, comment 1.

7. *The section of the Introduction from lines 60-97 could be shortened. In general, the Introduction is too long.*

The introduction was shortened, especially with respect to the lines 60-97.

Changed text: See Reviewer 1, comment 1.

8. *Lines 53-55: State here that sclerostin is a Wnt antagonist. Wnt binds to the co-receptors LRP5 (or LRP6) and Frizzled.*

We state now that sclerostin is a Wnt antagonist.

Changed text (Introduction, I.70 - 73):

Sclerostin antibodies (SclAB) inhibit a Wnt/ β -catenin-signaling antagonist protein, promoting osteoblast activity and suppressing osteoclast activity¹². Wnt/ β -catenin-signaling is initiated when a Wnt ligand binds to a Frizzled family receptor and its co-receptor LRP5/6 on the surface of a cell, resulting in β -catenin stabilization and target gene transcription inside the cell¹¹.

9. *Lines 75-78: The mention of microdamage associated with bisphosphonates requires supporting citations. This is an important issue, albeit with limited clinical data.*

The introduction was shortened and this section was deleted. In the discussion, the reference was inserted:

Changed text (Discussion, I.424 - 426): Regions with high local strain are linked to higher occurrences of microscopic tissue damage³⁸ of which even small amounts can lead to large reductions in strength³⁹ and increase fracture risk⁴⁰.

10. *Results*

'Dependence of (re)modeling probability on mechanical stimulus' and 'Synergisms in (re)modeling velocity and mechanostat parameters' sections could benefit from more descriptive headers and clear summaries or key takeaways at the end of each section to help guide the reader through the findings.

Changed headers: See Reviewer 2, comment 5.

Changed text (Results, I.319 - 321): These findings suggest that a reduction in RmT may be a prerequisite for strength gains; however, actual increases in strength require anabolic potential (Fig. 5e,f).

Changed text (Results, I.346 - 350): Bisphosphonates, alone or with mechanical loading, help preserve trabecular interconnectedness. Anabolic treatments combined with mechanical loading effectively enhance predicted strength by promoting formation and suppressing resorption. These additive effects across parameters likely underlie the observed synergistic gains in predicted strength, at least in SclAB.

11. *In 'Dependence of (re)modeling probability on mechanical stimulus' section, discussion about Fig. 5e, CCR is missing. Additionally, the implication of changes in non-targeted formation and resorption (Fig. 5i and j) should be discussed in the result. Many readers will not understand what you mean by "non-targeted formation/resorption".*

Given the feedback of Reviewers 2 and 3, we have substantially revised Figures 4 - 6. Non-targeted formation/resorption is not used any more in the manuscript. In Fig. 4d), min_R provides

now the information that could be connected with non-targeted resorption, but on a more conceptual level. Furthermore, we have inserted a description of CCR.

Changed text (Results, I.272 - 277): "Combined anabolic treatments resulted in higher correct classification rate (CCR) values compared to VEH and the BIS groups (Fig. 4e), with CCR being a metric used to evaluate the accuracy of classifying local bone formation or resorption events based on predicted mechanical signals. However, CCR only correlated moderately with the increases in strength, indicating increased mechanosensitivity of the system may not fully explain the ability of bone to increase strength (Fig. 4f).

Changed text (Results, I.265 - 268): "The asymptotic resorption probability value \min_R , which can be interpreted as the basal level of resorption independent from the mechanical signal, was significantly lower in ML, PTH + ML, and in SclAB + ML than in VEH ($p < 0.05$, Fig. 4d, Table S.8).

12. Lines 127-131 and Fig 1d,e: The decreases in Tb.N and Conn.D for most groups are surprising to me. Please add some explanation or context. (Is the increase in BV/TV due solely to increased Tb.Th, and despite fewer trabeculae?)

We inserted the following text as explanation.

Changed text (Results, I.118 - 123): As previously described¹⁸, mechanical loading increases BV/TV mainly through trabecular thickening despite a reduction in trabecular number, explainable by a full loss of thin trabeculae - a pattern also seen in the SclAB-group (Fig. 1c,d). In contrast, PTH increased BV/TV through moderate thickening and sustained preservation of trabecular number, resulting in a smaller loss of connectivity. BIS maintained BV/TV by preserving existing trabeculae, including thinner ones, with minimal thickening (Fig. 1, b-e).

13. Figure 1b-e. The y-axis values do not make sense. Should 100% be 0%, ie., representing change from week 20 (start of treatment)?

Figure 1-f represents now the change from week 20, with 0% denoting no change from start of treatment.

Changed: Figure 1

14. Fig 1b-e. In addition to between group comparisons noted by 'o', consider adding a symbol indicating whether the delta for each group is different from zero (i.e., the treatment increase (or decrease) the parameter).

We have now inserted a symbol (#) to indicate when the delta for each group was different from zero.

Changed text: Figure legend 1: # indicates that the delta of the group is different from zero, as tested with t-test with null-hypothesis.

15. Lines 139-145: Please state how Eroded Surface is defined.

Inserted text (Results, l.139 - 142): Eroded surface **per bone surface** (ES), **reflecting the proportion of bone surface previously affected by resorption,** was significantly **lower in ML and SclAB compared to VEH, BIS, and PTH (p<0.05, Fig. 11).**

Changed Figure 1 legend: (i) mineralizing surface **per bone surface** MS (j) bone resorption rate **per bone volume** BRR (k) mineral resorption rate MRR (l) eroded surface **per bone surface** ES. **Significant differences are depicted with a symbol in the color of the corresponding group (o: to monotreatment, ●: to combined treatment, p<0.05).**

Changed figure 3 legend: (c) mineralizing surface **per bone surface** (MS) (d) bone resorption rate per bone volume (BRR) (e) mineral resorption rate (MRR) (f) eroded surface **per bone surface** (ES).

16. Section starting with line 162 and Figure 2: The comparisons of combination groups vs. the sum of the monotherapies to infer Interfering, Additive or Synergistic effects is an excellent concept, but it is unclear how this is implemented and additional justification should be stated in the Results. Is the height of each bar the mean value for that group? It is unclear if these comparisons can be supported with statistical testing to increase rigor. Also state in Figure what the time interval used is for the data in the bar graphs.

The same concern was also raised by Reviewer 2 (Comment 1). In response, we have reanalyzed synergy and antagonism by statistically using the interaction term treatment:loading of a linear model. In addition, we describe now in the figure legends what the height of each bar represents. We also insert the time interval used for the data in the bar graphs. In addition, we provide all time intervals in the Supplementary Material.

Changed text (statistical testing for synergy): See Reviewer 2, comment 1.

Changed figure legends 2-6: **Bars show group means of sum of single vs. combined treatment indicating: antagonistic (I), additive (A) or synergistic (S) effects determined via treatment:loading interaction in a linear model on the ...**

Changed figures: 1-6: now stating the time intervals in the y-axis titles.

17. Lines 184- : The measures of trabecular "strength" estimated/predicted using FEA, and should be referred to as 'Predicted strength'.

The word "strength" has been replaced by "predicted strength" throughout the manuscript.

18. Line 189: Suggest that you add a statement explaining what is meant by an 'interfering' effect, and how you interpret this.

Thank you for pointing out the need for clarification. We now include a more explicit explanation of what is meant by synergy and antagonism. The word "interfering" was replaced with "antagonistic".

Added text (Methods, l.593 - 600): **"A synergistic effect was defined as a combined treatment response that exceeded the sum of the individual effects of treatment and mechanical loading. This implies that the two interventions may act through the same or facilitating biological pathways.**

Statistically, such synergy is indicated by a significant and positive interaction term between the main effect (treatment) and mechanical loading. An antagonistic effect was defined as a combined treatment response that was less than the expected additive effect of treatment and mechanical loading. Statistically, such antagonism was reflected by a significant and negative interaction term between the main effect (treatment) and mechanical loading.

19. *Figure 2j not referenced in the text.*

Figure 2j (now Figure 2g) is now referenced in the text.

Changed text (Results, I.185 - 187): Illustrating the effect of decreasing SED, the visualization of the 6th caudal vertebra in a ScIAB + ML animal shows a highly loaded trabecula at week 20, followed by structural reorganization and reduced SED at week 24 (see Fig. 2g).

20. *Line 214: Lowered compared to what?*

Changed text (Results, I.137): As expected, mineral resorption rate (MRR) was significantly lower in the BIS-group compared to VEH and...

Changed text (Results, I.141): ...was significantly lower in ML and ScIAB compared to VEH, BIS, and PTH ($p < 0.05$, Fig. 1l).

Deleted text (Results, I.214): "It was lowered in the ScIAB group..."

21. *Line 215: Is "beneficial" correct? Graph looks like Interfering.*

Has been replaced with "antagonistic" according to the above definition of antagonistic effect.

22. *Line 224: Typo "inlets" à "insets".*

The word has been replaced to "insets".

23. *Fig 3h: the second graph shows 'stderr' rather than R^2.*

We thank the reviewer for pointing out the confusion regarding the use of standard error. Our intention was to distinguish between linear and non-linear regression analyses, where R^2 is not always appropriate. To clarify this, we have revised the figures to display Pearson's R for linear regressions and Spearman's correlation coefficient ρ for non-linear regressions based on \log_2 -transformed models.

Changed figures: 2h, 3h, 6b, 6c

Changed text (Methods, I.625 - 630): Linear correlations between the %-change in predicted strength and parameter outcomes were assessed using Pearson's correlation coefficient R. For nonlinear relationships, we employed a power-law model of the form $y = a \cdot x^b$, which was linearized via logarithmic transformation to $\ln(y) = \ln(a) + b \cdot \ln(x)$. Parameters a and b were estimated using linear

regression on the transformed data, and the Spearman's correlation coefficient ρ between the log-transformed observed and predicted values was used to assess the goodness of fit.

24. Line 240: Word "ranges" is vague here.

The word "ranges" in "... formation and resorption occur in spatially distinct ranges." was replaced with "strain windows" (l.233 and l.243).

25. Lines 250, 252: The term "natural law" is an overstatement. Please find another way to state this.

Deleted text:

p.13, l.252 (old; Results): "~~of this apparent natural law~~".

p.13, l.254 (old; Results): "~~It could, however, be that the natural law is that strong that it masks subtle differences between treatments.~~"

26. Figure 4: It is unclear what %diff SED is showing; it is explained in the text, but could be also stated in the figure caption. It looks like the average value of %diff SED for quiescent group is not always equal to 0%. If so, can you explain how this can be.

Changed Figure 4 legend: **(b) SED at formation, resorption, and quiescent sites⁶, normalized to the group mean of quiescence to allow statistical testing with this group.**

27. Section starting on Line 260 Dependence of (re)modeling probability on mechanical stimulus. This is one of the most interesting aspects of the study, but the presentation is challenging as it lacks quantitative values (Figs 5a-d) and is unclear if statistical testing can be performed to support the comparison of AUC values and probabilities of non-targeted formation and values of RQ and QF.

We acknowledge that the presentation of Figure 5 was challenging to interpret. We have reframed Figures 4 and 5 and present now slightly other parameters to describe mechanoregulation and its implications on bone strength. We follow the publication of Marques et al. (2023) who formally assessed the ability of time-lapsed micro-computed tomography scans and finite-element-analysis to extract mechanostat parameters from experimental data. Furthermore, as this reviewer suggested, we quantify and test statistically, the coefficients of the fits of the conditional probability curves. Min_R represents non-targeted resorption, CCR the amount of mechanoregulated (re)modeling, summarizing AUC_R and AUC_F in one variable. The (re)modeling threshold is a representation of the intersection values RQ and QF. With this, we hope that the presentation of Figure 5 becomes clearer.

Changed Figure 4 legend: Figure 2: Formation aligns with high mechanical signal, resorption with low: (a) Spatial comparison of local mechanical signal (SED at week 20) with sites of trabecular bone formation and resorption (w20-22). (b) SED at formation, resorption, and quiescent sites⁶.

normalized to the group mean of quiescence to allow statistical testing with this group. The box spans the interquartile range (IQR) from the 25th to the 75th percentile. The median is shown as a line inside the box. Whiskers extend to the most extreme data points within $1.5 \times \text{IQR}$, points beyond this range are plotted as outliers. * $p < 0.05$, *** $p < 0.001$ using two-sided paired Wilcoxon-signed rank test with Bonferroni correction. (e) Conditional probabilities of formation and resorption as a function of SED. VEH indicated by black dotted line. Values are cropped above 40% due to low voxel count⁶. (d) Asymptotic resorption probability value (min_R). Δ denotes change from VEH; sum of single vs. combined treatment group means indicating antagonistic (I), additive (A) or synergistic (S) effects (e) CCR as a measure of mechanoregulated (re)modeling (f) Linear regression analysis in interval 20-22 of Δ Strength with CCR. Significant differences are depicted with a symbol in the color of the corresponding group (o: to monotreatment, ●: to combined treatment, $p < 0.05$).

Figure 5: (Re)modeling velocity curves as an approximation of the mechanostat theory. (a) (re)modeling velocity (RmV) curves for VEH / ML, BIS / BIS + ML, PTH / PTH + ML, SclAB / SclAB + ML; (b) formation saturation level (FSL) (c) resorption saturation level (RSL), (d) (re)modeling threshold (RmT) (b-d): The box spans the interquartile range (IQR) from the 25th to the 75th percentile. The median is shown as a line inside the box. Whiskers extend to the most extreme data points within $1.5 \times \text{IQR}$, points beyond this range are plotted as outliers. Δ denotes difference from VEH as no baseline value from week 20 was available. Bars show group means of sum of single vs. combined treatment indicating: antagonistic (I), additive (A) or synergistic (S) effects determined via treatment:loading interaction in a linear model on the absolute values. Significances were calculated using ANOVA followed by Tukey's post-hoc test; significant differences are depicted with a symbol in the color of the corresponding group (o: to monotreatment, ●: to combined treatment, $p < 0.05$). (g) Linear regression of Δ Strength with FSL, and power-law regression analysis with RmT from week interval w20-22.

28. Figure 5a-d: Include equations of these exponential curve fits in Supplemental. Do constants of these eqns have meaning that can support the qualitative descriptions of the curves in the text (e.g. Line 270-271). In general, the qualitative descriptions and the statements referring to steeper slopes, etc, lacks quantitative rigor. Authors should mention the slope values to give the readers the opportunity to better appreciate the magnitude of differences in mechanical sensitivity among the groups.

Supplementary material contains now a table for the coefficients derived from the conditional probabilities of formation and resorption. The curves were fitted per animal which allows statistical testing of the resulting coefficients. The asymptotic value of the conditional probability for bone resorption is used to assess the term called before "non-targeted remodeling" and is presented in Fig. 4d.

Changed Figure 4 legend: see comment 27.

Changed text (Results, I.261 - 268): "The probability for bone formation increased with increasing SED, and the probability for bone resorption decreased with increasing SED (Fig. 4c, Supplementary material Fig. S.3c). Especially the PTH + ML and SclAB + ML conditional probability curves opened up compared to their monotreatments, indicating a higher responsiveness to mechanical signals (Fig. 4c). The asymptotic resorption probability value min_R , which can be interpreted as the basal level of resorption independent from the mechanical signal, was significantly lower in ML, PTH + ML, and in

ScIAB + ML than in VEH ($p < 0.05$, Fig. 4d, Table S.8). Further comparisons of the coefficients from fitted functions to the conditional probabilities are provided in Supplementary Tables S.7 and S.8.

29. *Figure 5e,f,g are not cited in the text.*

Figure 5 has been rearranged. All subpanels are now cited in the text.

30. *Lines 276-284: If you want to include the AUC values, it would be better in a table than in the text.*

We don't show any more the AUC values. All parameter values are now presented in the Supplementary Material.

31. *Line 285: Suggest revise to state "In ML, the probabilities of non-targeted formation..."*

The text regarding non-targeted formation has been deleted to alleviate understandability of the last two figures (see Reviewer 2, comment 1 and Reviewer 3, comment 27).

32. *Fig 6f-h. Unclear that these RmV curves are relative to the VEH curve. State in the caption.*

The RmV curve comparisons have been deleted to improve understandability of Figure 6 (see Reviewer 2, comment 1). Instead, we use now consistently the graphs showing the sum of monotreatments compared to the combination treatments.

33. *Figure 6 and related text (lines 349-357): Values of RmV seem very low (eg, 0.00585 um/day). Are these correct? Why so different from histological mineral apposition rates which are on order of 1 um/day?*

Thank you for pointing this out. In fact, the values had to be rescaled by a factor of 1000 to convert into the right unit [kPa]. We have now corrected this.

Changed Figure: 6

34. *The implication of changes in RSL and FSL (Fig. 6i and j) should be discussed.*

Added text (Results, I.299 - 310): FSL was increased in all treatments compared to VEH ($p < 0.001$, Fig. 5b, Supplementary Table S.9), and was significantly higher in anabolic combination groups compared to the respective monotreatments ($p < 0.001$). This suggests an increased mechanosensitivity of the cells involved in bone formation. RSL was increased in the BIS and all combination groups compared to VEH ($p < 0.05$, Fig. 5c). The increase in RSL under bisphosphonate treatment may be explained by the suppression of bone turnover: at comparable strain levels, resorption would typically occur, resulting in a lower RSL; however, due to inhibited osteoclast activity, this resorptive response is blunted, making the system appear saturated at a higher RSL value. Regarding the role of additional mechanical loading, only the PTH + ML group showed a significant increase compared to PTH alone ($p < 0.05$, Supplementary

Table S.9). This suggests that, under strain conditions that would typically lead to bone resorption, the balance has shifted towards bone formation.

35. Discussion

Although the BIS and BIS+ML groups exhibit the lowest RmT values, suggesting a lower threshold for remodeling, they also demonstrated the least overall response. This apparent discrepancy should be addressed in the discussion section.

Added text (Results, I.310 - 321): RmT values, representing the setpoints in the mechanostat theory, were decreased in all pharmacological treatment groups compared to VEH ($p < 0.05$), with no significant improvements observed in the combination groups receiving mechanical loading (Fig. 5d). These results were even more pronounced in the second time interval (Supplementary Fig. 4d, Table S.9). The reduction in RmT suggests an upregulation of osteocytic mechanosensitivity, an expected outcome in the anabolic treatment groups, but an unexpected finding in the bisphosphonate-treated groups. This may reflect a compensatory osteocytic response to the suppressed bone resorption induced by bisphosphonate treatment. Furthermore, FSL correlated strongly with predicted strength increases ($R = 0.888$), while RmT exhibited a moderate inverse correlation ($\rho = 0.609$) when fitted with a power-law function.

36. Fig 5a-d: Since the differences in conditional probabilities between the monotreatment and the loading + treatment groups are relatively small (approximately 1–2%), the authors should consider commenting on the biological or practical significance of these differences in the discussion section.

As we provide in the revision also quantitative data to assess differences in conditional probabilities, it is now even easier to comment on the biological significance of these differences. We have added this to the results section where the findings are presented.

Changed text (Results, I.268 - 272): Further comparisons of the coefficients from fitted functions to the conditional probabilities are provided in Supplementary Tables S.7 and S.8. The presence of few significant differences between groups suggests mechanosensitivity of the system is largely preserved regardless of the presence or absence of mechanical loading.

Changed text: (Acknowledgement): "...and Júlia van den Nest Molina have supported the analysis of [...] and mechanoregulation mechanisms."

37. Line 370-71: "this is regulated via the mechanostat" – can you support this with citations, or qualify it "proposed to be regulated via the mechanostat". This study is the most convincing I have seen to support that this is true.

We have reformulated to "proposed to be regulated via the mechanostat" and inserted citations.

Changed text (Discussion, I.371): ...proposed to be regulated via the mechanostat^{6,19-21}

38. Line 402: "46-bladed propellers" seems typo. Usually described as "4-beta propeller structures"

Renamed to "four beta-propeller domains" but then deleted in the latest version. Thank you.

39. The SED maps in Fig. 5m show a reduction in SED over time. Given that the manuscript argues higher SED is associated with increased bone formation, does this observed reduction imply a decrease in bone formation capacity over time?

Thank you for your insightful comment. The observed reduction in Strain Energy Density (SED) over time in Fig. 5m (now: Figure 6a) does not necessarily imply a decrease in bone formation capacity. Rather, it reflects the dynamic nature of the mechanical loading and treatment effects, which contribute to a saturation window. Specifically, reductions in SED during the loading and treatment periods are consistent with the model of mechanostat theory, where bone formation is responsive to mechanical stimuli up to a certain threshold. Once this threshold is reached, the system enters a saturation phase, where the rate of bone formation stabilizes despite continued mechanical loading. This plateau is a natural result of the system's regulatory mechanisms, which aim to maintain bone homeostasis and prevent over-formation, rather than a decline in bone formation capacity. Thus, the reduction in SED is indicative of the system stabilizing within this saturation window, rather than a loss of formation potential over time.

We hope this clarifies the relationship between SED and bone formation in the context of mechanical loading and treatment effects.

Changed text (Discussion, I.424 - 428): Regions with high local strain are linked to higher occurrences of microscopic tissue damage³⁸ of which even small amounts can lead to large reductions in strength³⁹ and increase fracture risk⁴⁰. Reductions in strain energy density (SED) over time do not necessarily imply a decrease in bone formation capacity but rather reflect system saturation, consistent with mechanostat theory, where bone formation plateaus once strain thresholds are reached.

40. Methods

Lines 552-554: The method for bone strength estimation seems very sound, but is based on assumptions about failure criterion that were developed for human bones (Pistoia #59). Have these been shown accurate for estimation of mouse vertebral strength?

We included the following reference (Nyman et al. 2015):

Changed text (Methods, I.553): "This criterion has been shown applicable to the mouse vertebra as well⁵².

41. Line 577: Define CCR.

Changed text (Results, I.574): Correct classification rate (CCR) values were calculated from the mean conditional probabilities from all animals, returning a scalar value between 0.33 and 1 assessing

overall mechanoregulation, with 0.33 meaning (re)modeling is independent of the mechanical signal and 1 meaning it is entirely dependent on the mechanical signal.

42. *Lines 607-...: Are there any citations to support the definitions of Synergism, Additivity and Interference? I think these are very clear and reasonable, but perhaps there is supporting literature on this?*

Please see Reviewer 2, comment 1 who has requested to use another concept of assessing synergy, additivity or interference using statistical methods.

43. *The mechanical loading protocol should be described in greater detail. Additionally, did the applied 8N supraphysiological load induce any microdamage in the bone? What is the failure load of the 6th caudal vertebra?*

The mechanical loading protocol is now described in greater detail. We have added an explanation regarding the occurrence of microdamage and predicted failure load.

Added text (Methods, I.491 - 504):

Mechanical loading

A validated *in vivo* mouse tail loading model was used to induce trabecular bone adaptation under controlled mechanical conditions⁴³⁻⁴⁵. Stainless steel pins were implanted into adjacent vertebrae under fluoroscopic guidance three weeks prior to loading. An 8N cyclic axial load (3000 cycles, 10 Hz) was applied to the 6th caudal vertebra (CV6) using a custom loading device (caudal vertebra axial loading device, CVAD⁴⁴) three times a week. Control animals (0N groups) underwent identical procedures, including pin placement and anesthesia but no load was applied. A force of 8 N was used in order to maximize the effect within a short timeframe. It was the maximum force that could be applied based on what the pins within the bone could bear, and represents twice the force exerted on mouse vertebrae under physiological conditions⁴⁶. No microdamage has been observed to date, although not systematically assessed (unpublished data). In our experience, mechanical limitation arises at the pin sites due to concentrated forces, not within the loaded vertebra. Pistoia simulations estimate a maximum failure load of 24.5 N at CV6 in osteoporotic mice at week 20 (predicted trabecular strength, see Table S.2).

44. *In 'Pharmaceutical treatment' section, the delivery method for SOSTab is not specified and should be included.*

Changed text (Discussion, I.512 - 513): ..., *i.v. weekly into the lateral tail vein.*

45. *Figures – additional comments*

The use of 'o' in the color of corresponding group to denote statistical significance is innovative; however, it only represents $p < 0.05$ and the 'o's are very small. If there are results with more stringent significance levels (e.g., $p < 0.001$), the authors should consider denoting these separately to reflect the varying degrees of statistical strength, and consider making the symbols larger.

All statistical significances are now also reported in the Supplementary Material. We have there the possibility to show more stringent significance levels (e.g. $p < 0.001$). With the possibility to lookup the significance levels in the Supplementary Material, we hope it is not necessary any more to indicate separately in the figures. All symbols in Figures 1, 3, 5, and 6 have been made slightly larger.

46. The authors should present the data with individual data points for each mouse, where applicable, to visualize the variability within groups.

We have changed the figures to present the data with individual data points for each mouse to visualize the variability within groups.

Changed figures: 1-6

47. All figure captions should clearly state what the box-and-whisker plots represent, e.g., whether the box indicates the interquartile range, the line represents the median, and the whiskers show the full range or a specific percentile.

The figure captions now state more clearly what the box-and-whisker plots represent.

Changed Figure legends 1, 3- 6, S.1-S.4: The box spans the interquartile range (IQR) from the 25th to the 75th percentile. The median is shown as a line inside the box. Whiskers extend to the most extreme data points within $1.5 \times \text{IQR}$, points beyond this range are plotted as outliers.

48. Fig 5 is missing the color key for formation, resorption, and quiescence.

Fig. 5 (conditional probabilities) has been merged with Fig. 4 which now contains the color key for formation, resorption and quiescence.

49. Minor comments

The manuscript primarily discusses changes in trabecular bone; however, were any changes observed in the cortical shell of the vertebral body in response to mechanical loading and/or drug treatments?

We provide now values of the cortical shell in the Supplementary material to provide a more holistic picture of the changes occurring in the caudal vertebra (Table S.3, Figure S.1).

50. As Table 1 (list of statistical significances) does not provide any key additional information, this can be moved to the supplementary or consolidated with the figure. Also, a color key in the table will be helpful.

Table 1 has been deleted. A color key has been inserted in Supplementary Table S.1. Supplementary Tables S.2-S.9 yield now the absolute values and significance values in the Supplementary material.

Deleted Table: 1

Response to Reviewers (Manuscript: "Combined physical and pharmacological anabolic osteoporosis therapies synergistically increase bone response and local mechanoregulation")

Changed figure legends 1-6, S.2-S.4: (o: to monotreatment, ●: to combined treatment, see Supplementary Table S.1 for color encoding, $p < 0.05$)

Reviewer #4 (Remarks to the Author):

Thank you for your time and effort.

POLICIES AND FORMS REQUIRED FOR RESUBMISSION

1. *If the research findings apply to only one sex or gender, that must be indicated in the title and/or abstract. Information on the point should be included in the revised manuscript and detailed in the cover letter.*

We have inserted "female" as a word in the abstract.

2. ** Your paper uses custom code/software. Please complete the following code and software submission checklist and make your code available for reviewer assessment, if you have not already done so. The code/software can be provided in a zip file with a readme.txt file or other instructions for installing and running the software. If appropriate, also provide example data and expected output. If you have any issues with the file upload, please let me know.*

<https://www.nature.com/documents/nr-software-policy.pdf>

We provide the code in zenodo.org. The nr-software-policy.pdf is attached.

DATA AND CODE AVAILABILITY

3. ** All Nature Communications manuscripts must include a "Data Availability" section after the Methods section but before the References. If any of the data can only be shared on request or are subject to restrictions, please specify the reasons and explain how, when, and by whom the data can be accessed. For more information on this policy and a list of examples, see:*

<https://www.nature.com/documents/nr-data-availability-statements-data-citations.pdf>

We provide the data in zenodo.org. The nr-data-availability-statements-data-citations.pdf is attached.

4. ** Please also include a "Code Availability" section after the "Data Availability" section. If the code can only be shared on request, please specify the reasons. For more information on our code sharing policy and requirements, please see:*

<https://www.nature.com/nature-portfolio/editorial-policies/reporting-standards#availability-of-computer-code>

We provide a Code Availability section in the manuscript.

5. ** As Nature Portfolio policies strongly encourage you to share your research data in a public repository (e.g. spreadsheets, text, images), we are partnering with the figshare repository so that you can use the figshare integration via the 'Research Data Deposition' tab when submitting your revised manuscript. Data are stored privately until a manuscript decision is reached and you can edit/withdraw them up to this point: you retain rights and control over your data. The data will be published at the same time as your article; you will receive a data DOI, with guidance on linking the data and manuscript. In the event your manuscript is not accepted, you can keep or remove your data in figshare. We recommend the use of discipline-specific repositories where available and for a number of data types this is mandatory. Ensure you do not submit these data types or any sensitive data to figshare.*

** We strongly encourage you to deposit all new data associated with the paper in a persistent repository where they can be freely and enduringly accessed. We recommend submitting the data to discipline-specific and community-recognised repositories; a list of repositories is provided here: <http://www.nature.com/sdata/policies/repositories>. Refer to our data policies here: <https://www.nature.com/nature-portfolio/editorial-policies/reporting-standards#availability-of-data>*

** To maximise the reproducibility of research data, we ask that you provide a Source Data file containing the raw data underlying the following types of display items:*

- Any reported means/averages in box plots, bar charts, and tables*
- Dot plots/scatter plots, especially when there are overlapping points*
- Line graphs*
- Uncropped and unprocessed scans of all blots and gels including all quantified replicates. The edge of membranes, molecular weight ladders and loading controls should be presented on all blots. Where membranes have been cut, please ensure that at least one marker above and below is present. For an example of presentation of full scan blots, see the Source Data file of <https://www.nature.com/articles/s41467-020-16984-1#Sec35> and for more information, please refer to <https://www.nature.com/nature-research/editorial-policies/image-integrity>.*

The data should be provided in a single Excel file with data for each figure/table in a separate sheet, or in multiple labelled files within a zipped folder. Name this file or folder 'Source Data', and include a brief description in your cover letter. The "Data Availability" section should also include the statement "Source Data are provided with this paper." To learn more about our motivation behind this policy, please see: <https://www.nature.com/articles/s41467-018-06012-8>. A Source Data file is not necessary if all display items presented in the main manuscript and supplementary information can be reproduced from raw data and code that have already been shared in a public repository.

We provide the data and code in zenodo.org.

** Please replace any bar graphs with plots that feature information about the distribution of the underlying data. All data points should be shown for plots with a sample size less than 10. For larger sample sizes, please consider box-and-whisker or violin plots as alternatives. Measures of centrality, dispersion and/or error bars should be plotted and described in the figure legend.*

HOW TO SUBMIT

- Revised manuscript*
- Supplementary material*
- Point-by-point response to the reviewers' comments, reproduced verbatim*
- Cover letter to the editor*
- Author list changes: www.nature.com/documents/nr-author-list-change-form.pdf*
- Editorial policy checklist: <https://www.nature.com/documents/nr-editorial-policy-checklist.pdf>*

Response to Reviewers (Manuscript: "Combined physical and pharmacological anabolic osteoporosis therapies synergistically increase bone response and local mechanoregulation")

- *Reporting summary: <https://www.nature.com/documents/nr-reporting-summary.pdf>*
- *ORCID number corresponding author in the submission system; Please also inform all co-authors that they can add their ORCIDs to their accounts and that they must do so prior to acceptance. <https://mts-ncomms.nature.com/cgi-bin/main.plex?el=A1S2ELLS4A7DLvJ5l1A9ftdpEym3YHmRywqD5zz30wIGwZ>*